# CLIMB: Taming the LoRA Residency Cliff in Multi-LoRA Serving

**Haoran Zhang** [1]  **Zhiyu Liang** [1]  **Decheng Zuo** [1]  **Hongzhi Wang** [1]

## Abstract

Multi-tenant multi-LoRA serving multiplexes many LoRA adapters on a single GPU under high utilization, where most device memory is reserved for the KV cache, leaving only a small residency budget $K$ for adapters. In this regime, adapter readiness is atomic: if an adapter is not device-resident, the engine must perform a mandatory fetch, stalling shared execution and amplifying tail latency system-wide. With only $K$ residency slots, we identify a LoRA residency cliff: once the active adapter working set exceeds $K$, time-to-first-token (TTFT) tail latency can exhibit a congestion collapse rather than smooth degradation. To tame this cliff, we propose CLIMB, a minimal ingress controller that enforces feasibility-first admission by queueing non-resident adapters outside the engine, prioritizing critical (VIP) traffic, and rotating background adapters via round-robin. On a cliff-inducing workload, CLIMB averts collapse, reducing VIP TTFT p99 from $38.7\,\mathrm{s}$ to $13.1\,\mathrm{s}$ at matched throughput ($10.66\,\mathrm{rps}$) by keeping VIP engine latency near $0.13\,\mathrm{s}$ and shifting the residual tail into explicit ingress queueing. Overall, CLIMB shifts fetch-induced stalls from inside the engine to managed ingress queues, mitigating tail amplification without throughput loss in the evaluated settings.

## 1. Introduction

Multi-tenant *Multi-LoRA* inference is increasingly common. A single GPU serves many tasks where each request carries a distinct LoRA adapter (Hu et al., 2022; Sheng et al., 2023a; Wu et al., 2024). We consider latency-critical VIP requests and throughput-oriented BG requests; at high utilization, *tail latency* dominates (Dean & Barroso, 2013).

---
[1]Faculty of Computing, Harbin Institute of Technology, Harbin, China. Correspondence to: Hongzhi Wang <wangzh@hit.edu.cn>.

*Proceedings of the $43^{rd}$ International Conference on Machine Learning*, Seoul, South Korea. PMLR 306, 2026. Copyright 2026 by the author(s).

To maximize throughput, continuous-batching engines reserve most device memory for the KV cache, leaving only a tightly budgeted residual envelope for adapters (Kwon et al., 2023). Crucially, adapter residency is an all-or-nothing prerequisite (Li et al., 2024; Sheng et al., 2023a). That is, a request can execute only if its adapter is already *ready* (device-resident). Even with paging, a missing adapter still requires a mandatory fetch; under continuous batching, such a critical-path fetch can block co-batched requests (head-of-line), amplifying tail latency system-wide (Kwon et al., 2023; Sheng et al., 2023a; Li et al., 2024; Yu et al., 2022).

Together, tight device-memory budgeting and all-or-nothing readiness imply a discrete *residency budget $K$* (the max number of adapters simultaneously resident/ready on the device) (Sheng et al., 2023a). Throughout each run, $K$ is a configured serving contract (e.g., vLLM `max_loras`), not a runtime-varying estimate of instantaneous free memory after KV-cache allocation. Dynamic memory management may inform how a deployment provisions this budget; CLIMB then controls admission under the provisioned fixed $K$. Operationally, this means CLIMB gates which adapter identities enter the engine; it does not reallocate KV-cache memory or change $K$ during a run.

As the active working set $W_t$ of distinct adapters fluctuates with arrival patterns and exceeds $K$, the system stops degrading gracefully. Instead, it exhibits a nonlinear *LoRA residency cliff* that a brief feasibility violation ($|W_t| > K$) triggers engine-side stalls, and delayed progress further exacerbates queueing and churn. Existing mitigations are often post-hoc and reactive (e.g., eviction/prefetch inside the engine), where decisions near the boundary can precipitate abrupt tail blow-ups; static reservation avoids this but sacrifices throughput headroom (Chen et al., 2024; Sheng et al., 2023a; Wu et al., 2024; Dean & Barroso, 2013). The core challenge is online feasibility: demand shifts at request granularity, and any miss that slips onto the engine's critical path can stall an entire batch. This motivates enforcing the budget *before dispatch* at ingress, rather than reacting inside the engine. This work asks: *under high utilization*, can we keep the system in the safe regime by making the residency budget an explicit constraint, without conservative reservation?

**Our approach.** We propose CLIMB, a minimal ingress

controller designed to tame the residency cliff. The core idea is *feasibility first*. That is, rather than allowing feasibility violations ($|W_t| > K$) to enter the engine, CLIMB enforces a hard gate at ingress. It effectively *moves unavoidable waiting out of the engine*, converting unpredictable fetch-induced stalls on the critical path into explicit, managed queueing. When the demand exceeds the budget $K$, CLIMB prioritizes VIP adapters and rotates BG adapters via a simple round-robin policy over the residual slots. We avoid prediction-heavy prefetching and in-engine heuristics, which cannot prevent on-demand misses near the $K$ boundary and thus still place adapter loads on the engine's critical path; CLIMB instead enforces feasibility at ingress.

**Key results.** On an extreme cliff-inducing workload ($W{=}8$, $K{=}4$), CLIMB reduces VIP TTFT p99 from 38.7 s to 13.1 s at matched throughput (10.66 rps; Table 1). A TTFT decomposition shows CLIMB keeps VIP engine p99 near the safe-anchor level ($\approx 130$ ms) and shifts the remaining tail into explicit ingress queueing.

- We characterize the *LoRA residency cliff* and provide a mechanistic diagnostic (TTFT decomposition and an interpretable knee law) around feasibility.
- We propose CLIMB, a minimal feasibility-first ingress controller (hard gating + VIP preference + BG round-robin) that externalizes unavoidable delay as explicit queueing.
- We evaluate across $W/K$ sweeps, time-series probes, and a focused set of baselines under throughput-aware, load-matched controls, quantifying background liveness cost.

**Roadmap.** The rest of the paper proceeds as follows: Section 2 formalizes the model; Section 3 analyzes the cliff and knee; Section 4 presents CLIMB; Section 5 evaluates it; Section 6 discusses related work; and Section 7 concludes.

## 2. Background and Problem Formulation

We formalize the serving setting, the discrete adapter residency constraint, and the objective tackled by CLIMB.

**System Model.** We consider a multi-tenant service on a single GPU using a continuous-batching engine (Kwon et al., 2023; Yu et al., 2022). Requests arrive online. Each request $r$ carries a LoRA adapter identity $a(r)$ and a class label $c(r) \in \{\mathsf{VIP}, \mathsf{BG}\}$, where VIP is latency-critical and BG is throughput-oriented.

**Latency Decomposition (TTFT).** Let $t_{\mathrm{arr}}(r)$, $t_{\mathrm{disp}}(r)$, and $t_{\mathrm{ft}}(r)$ denote the ingress arrival time, the dispatch time into the engine, and the first-token time, respectively. We decompose time-to-first-token (TTFT) into an explicit ingress waiting component and an in-engine component (Agrawal

et al., 2024):

$$
\begin{aligned}
\mathrm{TTFT}(r) &\triangleq t_{\mathrm{ft}}(r) - t_{\mathrm{arr}}(r) \\
&= \underbrace{t_{\mathrm{disp}}(r) - t_{\mathrm{arr}}(r)}_{\mathrm{queue}(r)} + \underbrace{t_{\mathrm{ft}}(r) - t_{\mathrm{disp}}(r)}_{\mathrm{engine}(r)}. \quad (1)
\end{aligned}
$$

Here $\mathrm{queue}(r)$ is pre-dispatch waiting, and $\mathrm{engine}(r)$ is post-dispatch latency. Ingress control can only directly affect $\mathrm{queue}(r)$.

**Residency Budget and Readiness.** Throughput-oriented serving reserves most device memory for the KV cache, leaving only a small residual envelope for adapters (Kwon et al., 2023). At time $t$, let $R_t$ be the set of device-resident (*ready*) adapters; the engine enforces a LoRA-slot cap $|R_t| \leq K$ (at most $K$ adapters) (Sheng et al., 2023a). A request can execute only if its adapter is ready, i.e., $a(r) \in R_t$. If $a(r) \notin R_t$ at execution time, the system must fetch/load the missing adapter, incurring a miss penalty. Section 3 analyzes how such misses interact with continuous batching to amplify tail latency.

**Working Set and Feasibility.** Let $W_t$ denote the *active adapter working set* at time $t$, the set of distinct adapters with active demand (queued at ingress or in-flight). We write $|W_t|$ for its size (Denning, 1968). We call the system *feasible* if $|W_t| \leq K$; otherwise it is *infeasible*, meaning the full active set cannot be simultaneously resident and ready under budget $K$. We refer to the resulting tail regime shift under high utilization as the *LoRA residency cliff*. When the time index is clear, we write $W \triangleq |W_t|$ for the working-set size. Because the feasibility boundary is defined relative to $K$, we separate capacity provisioning from admission control. Within a run, $K$ is set by the serving memory budget rather than by the admission policy. Allocating more adapter-residency capacity raises this budget and shifts the feasibility boundary to larger working sets; at any fixed budget, however, over-capacity demand creates a residency conflict. Operationally, admission control regulates the engine-admitted subset by delaying surplus adapters at ingress, rather than changing $K$ during the run.

**Problem Statement.** Given an online arrival stream and a fixed per-run residency budget $K$, we seek an admission policy that decides which requests/adapters to dispatch into the engine to minimize VIP tail TTFT, while maintaining high throughput and ensuring BG makes progress. Importantly, we aim to improve VIP tails *without buying performance via load shedding* (Welsh et al., 2001); instead, the policy may defer work at ingress so that non-resident slow paths do not enter the engine at inopportune times.

## 3. The LoRA Residency Cliff

Section 2 defines Multi-LoRA serving and the discrete residency budget $K$. When the active working set violates

feasibility, adapter misses become unavoidable and may fall on the engine's critical path. Under continuous batching, a critical-path stall can delay batch-iteration boundaries, inflating TTFT for many co-batched requests and producing a tail cliff; this motivates feasibility-first admission (Section 4) (Yu et al., 2022; Li et al., 2024).

We use the TTFT decomposition in Eq. (1) (defined in Section 2) to localize tail sources by separating $\text{queue}(\cdot)$ and $\text{engine}(\cdot)$ contributions. This is necessary because batch-coupled in-engine stalls can dominate TTFT even when ingress queueing is negligible.

### 3.1. A Discrete Feasibility Boundary: Working Set vs. Residency Budget

Multi-LoRA inference is constrained by a discrete residency budget. As defined in Section 2, the engine maintains a resident adapter set $R_t$ with $|R_t| \leq K$, and execution obeys *readiness atomicity*. A request $r$ can run only if $a(r) \in R_t$. Thus, once feasibility is violated, misses are not avoidable accidents but a structural consequence of insufficient residency slots.

We measure near-term adapter demand by the active working set $W_t$ (Section 2), the distinct adapters with pending or in-flight demand in the ingress queues or engine. Feasibility is determined by whether the working set fits in the residency budget, i.e., $|W_t| \leq K$. If $|W_t| \leq K$, there exists a steady-state configuration that keeps all actively demanded adapters resident aside from true cold starts. Otherwise ($|W_t| > K$), feasibility is violated, i.e., at least one actively demanded adapter must be non-resident at any moment, so progress necessarily entails explicit waiting or on-demand loads. Because the constraint is discrete, crossing this boundary by even one adapter can abruptly change tail behavior.

### 3.2. From Inevitable Misses to a Tail Cliff under Continuous Batching

Building on the TTFT decomposition above, we now explain *why* TTFT inflation can be engine-dominated and cliff-like under continuous batching. The key transition is that a residency violation turns adapter misses from occasional cold-start events into a recurring steady-state risk. When active adapters fit within the residency budget, a miss can often be amortized after the adapter is loaded. Once $K < W$, however, the system cannot keep every active adapter resident at the same time; without an ingress gate, fetches for non-resident adapters can enter the engine's critical path. Under continuous batching, these fetches can delay batch-iteration progress, reducing effective service progress, increasing backlog, and leaving more adapters simultaneously active. This feedback loop is what turns a discrete residency violation into a tail-latency cliff.

The cliff is not simply "more tenants $\Rightarrow$ more work." It arises from two ingredients: (i) misses become *unavoidable* once feasibility is violated ($|W_t| > K$), and (ii) continuous batching couples many sequences to a shared execution cadence.

Under continuous batching, the engine advances in token-step iterations over a micro-batch. A single on-demand adapter fetch/load on the critical path delays an iteration boundary, so even requests whose adapters are already resident can inherit the stall. Thus, a small fraction of misses can inflate the *engine* component in the TTFT decomposition (Eq. (1)) for many requests (Yu et al., 2022).

Moreover, stalls and infeasibility reinforce each other under open-loop arrivals. Stalls reduce effective service progress, backlog grows, and more adapters remain simultaneously active, sustaining (or worsening) $|W_t| > K$. This positive feedback explains the knee-like regime change that once the system cannot cover the active working set, tail latency can collapse into a slow regime instead of degrading smoothly (Harchol-Balter, 2013).

Section 5 validates these signatures that the combined $W/K$ sweeps show an abrupt transition as feasibility flips (Fig. 2), and the phase-shift alignment localizes the amplification to the engine component under global FIFO and its migration to ingress queueing under CLIMB (Fig. 3).

### 3.3. A Simplified Queueing Model (and an Interpretable Knee Law)

We give a mechanism-driven, window-level diagnostic that links the cliff to feasibility pressure ($K < W$) under continuous batching. The goal of this abstraction is not to predict exact TTFT values. Instead, it gives a compact way to reason about when feasibility pressure starts consuming service slack: $\lambda$ captures offered load, $p_{\text{miss}}$ captures discrete residency pressure, and $\tau$ absorbs the effective cost of a miss including batching-cadence amplification.

**Window abstraction.** Fix a short window. Let $\lambda$ be the total (VIP+BG) arrival rate in the window, and $s_0$ be the baseline fast-regime VIP engine time measured at a safe anchor ($K \geq W$). We use VIP engine time as the response variable and model it as $S \approx s_0 + \tau X$, where $X \in \{0, 1\}$ indicates whether a non-resident slow path stalls the engine cadence in the window. Here $\tau$ is an *effective* stall penalty that absorbs cadence-coupled amplification.

**Feasibility-driven miss pressure and effective load.** Let $W_{\text{window}}$ be the number of distinct adapters that appear in the window (VIP+BG). When $K < W_{\text{window}}$, misses are a feasibility consequence. We proxy miss pressure by $p_{\text{miss}} \triangleq \max\left(0, 1 - \frac{K}{W_{\text{window}}}\right)$, yielding the effective load

$$\rho_{\text{eff}} \triangleq \lambda \cdot \left(s_0 + p_{\text{miss}} \cdot \tau\right). \tag{2}$$

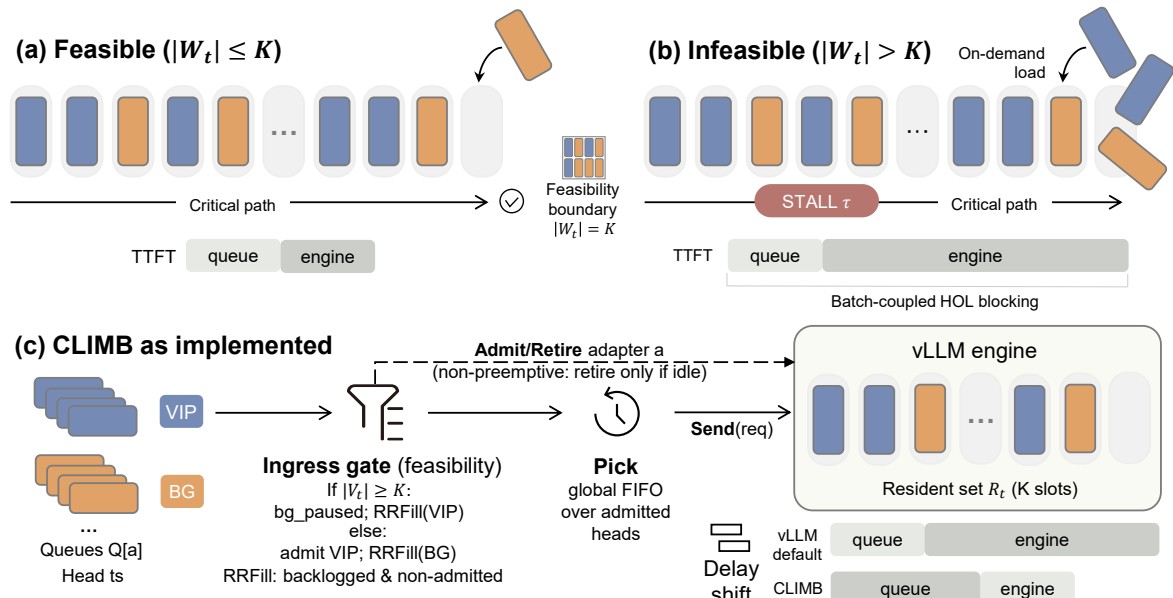

*Figure 1.* **Sec. 3/4 bridge overview.** The residency cliff is driven by discrete feasibility ($|W_t|$ vs. $K$) and cadence coupling under continuous batching. CLIMB mitigates the cliff by enforcing feasibility at ingress: non-resident adapters wait outside the engine, preventing fetch-induced engine stalls.

As $\rho_{\text{eff}}$ approaches one, even small feasibility violations ($p_{\text{miss}} > 0$) can consume slack and cause disproportionate tail growth, matching standard utilization blow-up intuition (Harchol-Balter, 2013).

**A knee law (informal).** This suggests interpreting the knee as the smallest $K$ that keeps $\rho_{\text{eff}}$ below a constant level (near 1). For $K < W_{\text{window}}$, substituting $p_{\text{miss}} \approx 1 - K/W_{\text{window}}$ into $\rho_{\text{eff}}$ and solving $\rho_{\text{eff}} \approx 1$ yields

$$K^{\star}(W_{\text{window}}) \approx W_{\text{window}}\left(1 - \frac{1/\lambda - s_0}{\tau}\right), \quad (3)$$

clipped to $[0, W_{\text{window}}]$. Thus the predicted knee shifts approximately linearly with working-set size, matching the near-linear knee movement in our $W/K$ sweeps. In stress runs with minimal slack ($1/\lambda \approx s_0$), Eq. (3) predicts $K^{\star} \approx W$.

**Proposition 3.1** (Feasibility-first admission externalizes unavoidable delay). *Suppose that an ingress controller maintains an admitted adapter set of size at most $K$ and dispatches only admitted adapters into the engine, while the engine keeps the admitted set device-resident in steady state (up to cold starts). Then admitted traffic has $p_{\text{miss}} \approx 0$ and hence operates near the fast regime ($S \approx s_0$); any excess demand necessarily manifests as explicit pre-dispatch queueing rather than amplified engine-side stalls.*

*Remark* 3.2 (Mild-overload dominance reversals at $W = K+1$). The avoided stall tail scales with miss pressure, on the order of $p_{\text{miss}}\tau$. Feasibility-first admission improves end-to-end VIP p99 only when the induced ad-mission/rotation queue tail is smaller than the avoided stall tail:

$$Q_{\text{VIP},p99}^{\text{climb}} - Q_{\text{VIP},p99}^{\text{fifo}} \lesssim p_{\text{miss}}\tau. \quad (4)$$

When $W = K+1$, $p_{\text{miss}} \approx 1/W$ is small, so (4) can fail, yielding a mild end-to-end reversal even if engine-side stalls drop sharply. We quantify this queue-side trade-off via VIP-absence intervals from controller logs (Appendix D.7).

**Scope and evidence.** Eq. (2) is a diagnostic rather than a literal end-to-end engine model. Appendix D provides proof sketches (including the knee implication) and ROC/AUC evidence that $\rho_{\text{eff}}$ separates "bad" versus "good" windows in the cliff regime.

### 3.4. Design Implication: Enforce Feasibility *Before* Dispatch

Because miss-induced slow paths that occur *inside* the engine can be amplified through continuous-batching cadence coupling, reacting only after requests are admitted is inherently late. A more robust mitigation is to act *at ingress*: when the active working set threatens to exceed $K$, explicitly restrict which adapters are eligible to dispatch and keep the rest waiting *outside* the engine. This externalizes unavoidable infeasibility as $\text{queue}(\cdot)$ rather than amplified $\text{engine}(\cdot)$ stalls, while preserving a stable engine cadence for admitted adapters (Welsh et al., 2001). Section 4 instantiates this feasibility-first principle as CLIMB.

# 4. CLIMB: A Minimal Feasibility-Biased Mitigation

Section 3 shows that the cliff is triggered when infeasible adapter demand ($|W_t| > K$) induces fetch stalls on the engine critical path, amplified by continuous-batching cadence coupling. CLIMB therefore enforces feasibility at ingress by gating dispatch eligibility, so unavoidable infeasibility appears as explicit ingress waiting (queue($\cdot$)) rather than amplified engine-side stalls (engine($\cdot$)) in Eq. (1), while leaving the engine's internal batching/dispatch unchanged.

## 4.1. Feasibility-First Ingress Gating (Control Surface and Contract)

Requests first enter per-adapter ingress FIFOs, which CLIMB assumes or wraps. CLIMB maintains an admitted set $A_t$ and dispatches only from the FIFOs of adapters in $A_t$, leaving non-admitted adapters queued and invisible to the engine; hence continuous batching, token scheduling, and in-engine ordering are unchanged, and CLIMB acts solely as an ingress eligibility gate (Welsh et al., 2001).

To make feasibility enforcement auditable, we explicitly track adapters in-flight in the engine. Let $I_t$ denote the set of adapters with at least one previously dispatched request not yet completed. CLIMB enforces the feasibility invariant on the effective active set

$$S_t \triangleq A_t \cup I_t, \qquad |S_t| \leq K, \tag{5}$$

where $K$ is the engine's discrete adapter residency budget (e.g., max_loras). Operationally, CLIMB admits a new adapter only when (5) has slack, and it releases a slot only when an admitted adapter becomes idle (its ingress queue drains and it has no in-flight requests). This non-preemptive rule avoids churn and ensures that feasibility is checkable at the controller boundary (Sheng et al., 2023a; Chen et al., 2024).

Under this contract, admitted traffic experiences near-zero miss pressure in steady state (up to true cold starts). The engine sees at most $K$ adapters, so miss-induced slow paths no longer enter the engine critical path. In turn, this directly instantiates Proposition 3.1: CLIMB externalizes unavoidable infeasibility into queue($\cdot$) rather than amplified engine($\cdot$) stalls.

## 4.2. VIP-Biased Admission with Round-Robin Filling

CLIMB biases feasibility toward latency-critical VIP traffic while preserving throughput and keeping the engine dispatch rule unchanged. Let $V_t$ be the set of VIP adapters with active demand (backlog at ingress or in-flight), and $B_t$ the analogous set for BG. When VIP demand oversubscribes the budget ($|V_t| \geq K$), feasibility is saturated by VIP demand. CLIMB pauses new BG admissions; paused

---

**Algorithm 1** CLIMB controller invoked on each dispatch attempt (skeleton).

---

**Require:** budget $K$; per-adapter FIFOs $Q[a]$; inflight set $I_t$; RR pointers $p_V, p_B$
**Invocation:** once per dispatch attempt;
    arrivals/completions update next-call state.
    **Ingress gating (Sec. 4.1)**
1: Remove from $A_t$ only adapters $a$ with $|Q[a]| = 0$ and $a \notin I_t$ ▷ idle-slot release
2: $W_t \leftarrow \{a : |Q[a]| > 0 \lor a \in I_t\}$; $V_t \leftarrow \{a \in W_t : \mathsf{VIP}\}$; $B_t \leftarrow \{a \in W_t : \mathsf{BG}\}$
    **Admission + RR filling (Sec. 4.2)**
3: **if** $|V_t| \geq K$ **then** ▷ bg_paused
4:     $A_t \leftarrow I_t \cup (A_t \cap V_t)$ ▷ drop non-inflight BG
5:     $A_t \leftarrow \mathrm{RRFILL}(A_t, V_t, K, p_V)$
6:     $E_t \leftarrow A_t \cap V_t$ ▷ dispatch VIP only
7: **else**
8:     $A_t \leftarrow A_t \cup I_t \cup V_t$
9:     $A_t \leftarrow \mathrm{RRFILL}(A_t, B_t, K, p_B)$
10:    $E_t \leftarrow A_t$
11: **end if**
12: Dispatch global FIFO over heads of $\{Q[a] \mid a \in E_t\}$

---

adapters stop receiving new dispatch eligibility, but requests already in flight are never preempted. Slot releases occur only when an admitted adapter becomes idle, and round-robin then rotates which VIP adapters occupy the $K$ slots. This provides best-effort liveness among VIP adapters while keeping the engine cadence stable for admitted VIP work. Otherwise, when VIP fits ($|V_t| < K$), CLIMB admits all VIP adapters in $V_t$ and fills the remaining $K - |V_t|$ slots by round-robin over BG adapters in $B_t$.

Crucially, VIP preference is applied at admission (which adapters are eligible), not via strict in-engine dispatch priority. Once an adapter is admitted, dispatch follows the same global FIFO rule as the baseline among heads of admitted queues. As discussed in Remark 3.2, this minimality can yield knife-edge reversals at $W = K+1$ when the avoided stall tail is smaller than the induced admission/rotation queue tail.

## 4.3. Algorithm

Algorithm 1 summarizes CLIMB. Arrivals and completions update the state observed by the next invocation; the controller decision routine itself is invoked at dispatch opportunities. At each invocation, CLIMB maintains an engine-visible admitted set $A_t$ whose effective resident proxy $A_t \cup I_t$ is capped by $K$, and dispatches only from adapters in $A_t$ (VIP-only when $|V_t| \geq K$), making feasibility an explicit invariant. The line structure of Algorithm 1 exposes three controller stages. Lines 1–2 release idle slots and rebuild active VIP/BG sets from queued plus in-flight work. Lines 3–11 choose the engine-visible adapters under $K$: the VIP-overload branch preserves in-flight work, revokes non-inflight BG eligibility, and exposes

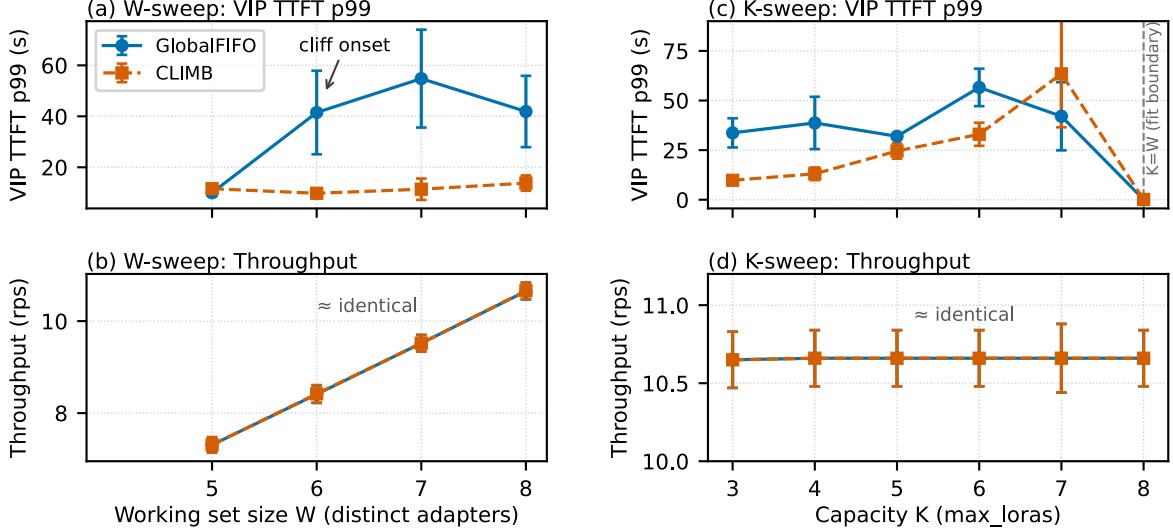

*Figure 2.* **Boundary and knee around feasibility via combined** $W/K$ **sweeps.** Left column: $W$-sweep at fixed $K{=}4$ with $W{=}5$–8. Right column: $K$-sweep at fixed $W{=}8$ with $K{=}3$–8. (a,c) VIP TTFT p99; (b,d) achieved throughput (rps). Curves compare GLOBALFIFO and CLIMB under the same open-loop arrivals (mean±std over 3 seeds). The cliff appears when $W > K$ (left) or $K < W$ (right) despite ≈ identical throughput, and the knee snaps back at the safe anchor $K = W$ (vertical dashed line).

only VIP queues, while the residual branch admits VIP and round-robin fills BG. Line 12 is the sole dispatch step, preserving global FIFO over admitted heads; thus CLIMB changes eligibility, not the engine's internal scheduling rule. RRFILL$(A, C, K, p)$ scans $C$ in round-robin order from pointer $p$, adding backlogged adapters ($|Q[a]| > 0$) not already in $A$ until $|A| = K$. Full implementation-aligned pseudocode and corner cases are in Appendix E. We explored richer variants (e.g., deficit-based fairness, stability guardrails, and dispatch-side DRR/rescue) (Shreedhar & Varghese, 1996); they mainly shift trade-offs and are summarized in Appendix F.

## 5. Evaluation

Section 3 predicts that the cliff arises when feasibility is violated ($|W_t| > K$) and miss-induced stalls are amplified by continuous-batching cadence coupling. We validate the boundary/knee, mechanism time-series (engine→queue migration), throughput-aware baseline trade-offs, and background liveness cost.

All experiments use open-loop, cliff-inducing stress workloads (Appendix A). For the fixed LoRA-slot budgets in our evaluation setup, each Qwen2.5-7B rank-128 adapter occupies about 0.301 GiB. A fully populated budget of $K{=}4$ or $K{=}8$ therefore corresponds to adapter-residency capacity of roughly 1.20 GiB or 2.41 GiB, respectively, not to instantaneous free-memory measurements. We report achieved throughput alongside tail latency to avoid closed-loop backpressure masking stalls (Jain, 1990; Agrawal et al., 2024). Our primary metric is VIP TTFT p99, and we also report

BG TTFT p99 and throughput (rps). App. Tab. 7 reports per-class rps and TokEq/s for throughput auditing. When diagnosing tail sources, we report p99 of queue$(\cdot)$ and engine$(\cdot)$ in Eq. (1) (percentiles are not additive). Unless stated otherwise, each point reports mean±std over three independent runs (seeds 101/102/103).

### 5.1. Boundary and Knee Around Feasibility: Combined $W/K$ Sweeps

Figure 2 consolidates two complementary sweeps that isolate the feasibility boundary between the resident-set budget $K$ and the active adapter working set $W$ under the same open-loop arrivals. The left column fixes $K{=}4$ and sweeps $W \in \{5, 6, 7, 8\}$, while the right column fixes $W{=}8$ and sweeps $K \in \{3, 4, 5, 6, 7, 8\}$. Across both sweeps, GLOBALFIFO (no admission control; App. Tab. 6) exhibits a sharp tail cliff once the system becomes over-capacity ($W > K$): VIP TTFT p99 inflates by orders of magnitude (Fig. 2a,c). Importantly, achieved throughput remains nearly unchanged across the two policies and across the sweep range (Fig. 2b,d), indicating that the cliff is not a throughput collapse but a latency blow-up driven by feasibility violation.

CLIMB mitigates this cliff without sacrificing throughput. In the stall-dominated regime (empirically, $K \le W - 2$ in our sweeps), CLIMB substantially reduces VIP tail latency at matched throughput, consistent with feasibility-first admission preventing miss-induced stalls from entering the engine critical path. As $K$ increases toward $W$, we observe a predictable knee: the system returns to the fast regime

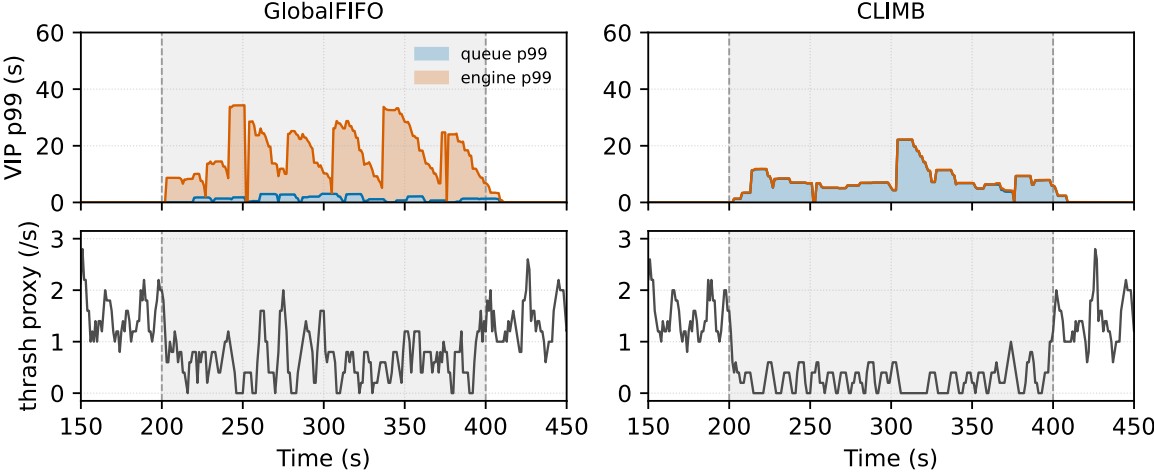

*Figure 3.* **Mechanism evidence via time-series alignment (phase-shift probe). Layout:** GLOBALFIFO (left) vs. CLIMB (right). **Top:** rolling VIP p99 of queue(·) and engine(·) (Eq. (1)), shown stacked only to indicate where tail mass concentrates (percentiles are not additive). **Bottom:** resident-set dynamics and switching activity (a proxy derived from resident-set change events). In over-capacity phases ($|W_t| > K$), GLOBALFIFO exhibits engine-dominated tail spikes; CLIMB collapses the engine component and shifts the residual tail into explicit ingress queueing.

and, at the safe anchor $K = W$, CLIMB becomes near-transparent and matches GLOBALFIFO (Fig. 2c). In our fixed-$W$ sweep, $K$ is an external capacity axis used to locate this boundary: increasing $K$ moves the operating point from deeper infeasibility toward the feasible side, while CLIMB controls admission at each fixed $K$. At the knife-edge $W = K + 1$ (visible at $W=5$ when $K=4$, and at $K=7$ when $W=8$), gains can diminish or mildly reverse, consistent with the mild-overload analysis in Section 3.3 (Remark 3.2): when stall risk is already small, the residual tail may be dominated by explicit ingress queueing rather than engine-side stalls.

### 5.2. Mechanism: Engine-Side Stall and Delay Migration

The sweeps above pin down *where* the feasibility boundary lies; we now show *how* the cliff manifests inside the system. We use a phase-shift probe (Appendix A.2) that alternates the system between a fit phase ($W \leq K$) and an over-capacity phase ($W > K$) under the *same* open-loop arrival process. Figure 3 aligns the resulting time series for GLOBALFIFO and CLIMB. In GLOBALFIFO, tail spikes appear immediately when entering the over-capacity phase and are dominated by *engine-side* delay, consistent with fetch-induced stalls occurring *after* admission and then being amplified by batching-cadence coupling. In CLIMB, the engine component stays near the safe-anchor level across phases, while the residual tail concentrates in *explicit queueing* at ingress. This engine→queue migration is the intended effect of feasibility-first admission: unavoidable waiting becomes visible and controllable, rather than an amplification-prone stall inside the continuous-batching engine.

*Table 1.* **Cliff vs. Safe regimes at fixed $W$=8.** VIP TTFT p99 decomposition and throughput (open-loop). Values are mean±std (3 seeds). Units are seconds (s) except throughput (rps). Queue/Engine columns report component-wise p99 and need not sum to Total p99.

| Policy | $K$ | VIP TTFT p99 | | | Thr |
|---|---|---|---|---|---|
| | | Total (s) | Queue (s) | Engine (s) | (rps) |
| GLOBALFIFO | 4 | 38.71±13.2 | 2.10±0.1 | 38.43±12.9 | 10.66±0.18 |
| CLIMB | 4 | **13.06**±3.2 | 12.97±3.2 | **0.13**±0.0 | 10.66±0.18 |
| GLOBALFIFO | 8 | 0.13±0.0 | 0.00±0.0 | 0.13±0.0 | 10.66±0.18 |
| CLIMB | 8 | 0.13±0.0 | 0.01±0.0 | 0.13±0.0 | 10.66±0.18 |

Consistently, the bottom panels show more bursty switching activity under GLOBALFIFO in the over-capacity interval, whereas CLIMB keeps switching activity comparatively steady.

To ground magnitudes, Table 1 reports two representative steady-state points at fixed $W$=8: a cliff point ($K$=4) and the safe anchor ($K$=8). At the cliff, GLOBALFIFO is engine-dominated (VIP engine p99 ≈ 38s), whereas CLIMB collapses the engine tail to the safe-anchor level (VIP engine p99 ≈ 0.13s) at matched throughput, shifting the remaining tail into queueing. At the safe anchor, both policies converge to the same fast regime, indicating near-transparency when feasibility is not at risk.

### 5.3. Trade-Off Across Baselines Under Throughput Awareness

We compare CLIMB against a small set of baselines that represent common ways to handle over-capacity adapter residency (Sheng et al., 2023a; Chen et al., 2024; Wu et al.,

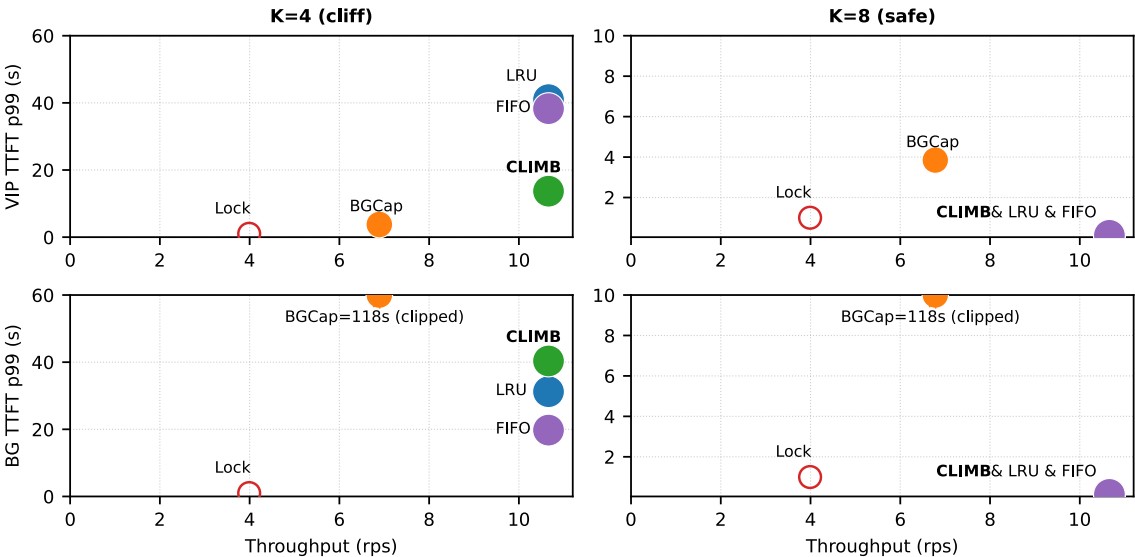

*Figure 4.* **Throughput-aware baseline positioning (fixed $W{=}8$).** We plot TTFT p99 versus achieved throughput (rps) to expose throughput confounding: a leftward shift indicates load shedding, while a downward shift at similar throughput indicates a genuine tail improvement at comparable load. Panels compare cliff ($K{=}4$) vs. safe ($K{=}8$) regimes, and VIP vs. BG tails. Legend abbreviations follow policy names (e.g.,FIFO=GLOBALFIFO, BGCap=BGCAP, LRU=LRUGATE, Lock=LOCKGATE).

2024): GLOBALFIFO (no admission control), BGCAP (static BG distinct-cap), LRUGATE (reactive locality-driven gating), and LOCKGATE (lock the active set to suppress switching). For readability, figure legends use abbreviated labels for these policies. (See App. Tab. 6 for a feature summary and implementation notes.)

A latency-only comparison can be misleading because some policies reduce tail p99 primarily by lowering achieved throughput and thus operating under a lighter effective load. To make this confounding explicit, Figure 4 plots tail TTFT p99 *against* achieved throughput (rps): a leftward shift indicates load shedding, while a downward shift at similar throughput indicates a genuine tail improvement at comparable load (Agrawal et al., 2024).

At the cliff point ($W{=}8, K{=}4$), GLOBALFIFO and CLIMB achieve similar throughput but diverge in tails: GLOBALFIFO keeps BG tails lower yet inflates VIP tails (e.g., $\sim$40 s versus $< 20$ s) via engine-side amplification under over-capacity, whereas CLIMB enforces feasibility-first admission to shift delay out of the engine, protecting VIP at the cost of higher BG tails. LRUGATE remains ineffective, leaving both classes with high tails at comparable throughput. BGCAP and LOCKGATE further reduce tails only with substantial throughput loss; LOCKGATE is the most throttling-dominated.

At the safe anchor ($W{=}8, K{=}8$), throughput-preserving policies converge to the same fast regime, and CLIMB is near-transparent, matching both latency and throughput

when feasibility is not at risk; any remaining differences primarily reflect intentional throughput reduction by more restrictive baselines.

### 5.4. Background Liveness Cost

Figure 4 reports BG tail latency (TTFT) under throughput awareness, but TTFT alone can understate starvation-like behavior when a policy intentionally queues requests before admission (Shreedhar & Varghese, 1996). We therefore quantify background *liveness* by measuring how long it takes a BG tenant to receive service once it becomes backlogged. For each BG adapter, define $t_{start}$ as the first time its backlog transitions from $0 \to> 0$ and $t_{served}$ as the first subsequent dispatch time; we define `backlogged_wait` $= t_{served} - t_{start}$. To avoid saturation and make this metric meaningful, we measure it under a reduced-load liveness probe at the cliff point ($W{=}8, K{=}4$).

Under this probe, the `backlogged_wait` p99 (overall / worst-adapter) increases from $2.73 \pm 0.1$ ms / $3.45 \pm 0.2$ ms under GLOBALFIFO to $5.44 \pm 0.75$ s / $4.54 \pm 0.63$ s under CLIMB (mean$\pm$std over 3 seeds), reflecting the fundamental BG liveness cost of keeping the engine feasible for VIP traffic. Here, *overall* p99 is request-weighted, while *worst-adapter* p99 takes the maximum per-adapter episode-weighted p99 (capturing the most unlucky tenant); because the two aggregates use different weights, the overall p99 can exceed the worst-adapter p99.

## 5.5. Additional Supporting Results

We highlight two compact supporting checks here before deferring the remaining details to the appendix. They are not new primary results: the sensitivity checks ask whether the VIP-tail ordering survives nearby workload changes, while a symmetric-prompt negative control asks whether the large gap weakens when the prompt-length asymmetry behind the residency cliff is removed.

For sensitivity, CLIMB shows the same qualitative VIP-tail ordering under several workload knobs. With shorter BG prompts (2048→1024 tokens), it reduces VIP p99 from about 42.9 s to 11.5 s at matched throughput (10.63–10.64 rps); across generation limits of 128/256/512 tokens, from 36.5–48.6 s to 16.7–18.4 s at 10.62–10.64 rps; and across additional VIP/BG mixtures, from 16.9–20.3 s to 7.85–10.3 s at 10.52–10.61 rps. Conversely, when VIP and BG prompt lengths are symmetric, the large GLOBALFIFO-versus-CLIMB gap largely disappears. This supports the interpretation that CLIMB's gains come from controlling residency-induced priority inversions near the cliff, consistent with the engine-to-queue migration in Figure 3, rather than from a generic engine-level speedup.

The remaining robustness results (rank sweep, second GPU), full baseline numbers and throughput-matched controls, expanded diagnostics/variants, and overhead appear in the appendix (Appendix A–I).

## 6. Related Work

**LLM serving and continuous batching.** Modern LLM serving systems improve throughput and latency via optimized attention kernels, KV-cache management, and dynamic/continuous batching (e.g., Dao et al., 2022; Dao, 2023). These works typically assume model weights are fully resident and focus on compute–memory trade-offs for tokens and the KV cache. In contrast, our setting introduces a small, discrete residency budget for adapter weights (LoRA modules) atop a large KV-cache footprint, where miss-induced stalls can be amplified by batching cadence into a tail cliff.

**Scheduling, admission control, and tail-SLO serving.** A broad line of work studies multi-tenant scheduling and admission control for tail-SLOs under shared resources (e.g., Welsh et al., 2001; Dean & Barroso, 2013). Most approaches assume per-request execution cost is stable once admitted; our cliff instead stems from a discrete feasibility constraint on adapter residency, where a miss can enter the engine critical path and propagate via continuous batching. CLIMB is intentionally minimal: it preserves the engine's internal dispatch order and gates only at ingress to enforce feasibility, turning amplified engine stalls into explicit and controllable queueing.

**Parameter-efficient fine-tuning and adapter management.** LoRA and related parameter-efficient fine-tuning techniques enable per-tenant specialization with small adapter weights (e.g., Houlsby et al., 2019; Li & Liang, 2021). Recent serving systems support dynamic adapter loading and multi-adapter execution (e.g., Sheng et al., 2023a; Chen et al., 2024; Wu et al., 2024). Our contribution is complementary: we characterize a feasibility-driven tail cliff when the active adapter working set exceeds a small residency budget, and propose a minimal feasibility-first mitigation with quantified trade-offs. Extended related work and broader context appear in Appendix J.

## 7. Conclusion

We identified a *LoRA residency cliff* in multi-adapter LLM serving: once the active adapter working set exceeds a small residency budget ($|W_t| > K$), misses become unavoidable and can be amplified by continuous batching into severe tail latency blow-ups without a throughput collapse. We presented a mechanistic diagnostic that explains the knee near feasibility and the mild-overload reversals at $W = K+1$, and introduced CLIMB, a minimal ingress controller that enforces feasibility before dispatch and externalizes unavoidable delay as explicit queueing. Across boundary sweeps, time-series alignment, and baseline comparisons under throughput awareness, CLIMB substantially reduces VIP tail latency at matched throughput, while incurring a quantified background liveness cost. Future work could extend this fixed-budget admission primitive with transition-aware admission that accounts for adapter-set turnover costs, while workload-aware provisioning tools could help choose the residency budget $K$ for a deployment without blurring the separation between provisioning choices and admission control under a chosen budget. Overall, we show that discrete feasibility constraints—not just offered load—can dominate tail behavior in LLM serving, motivating feasibility-first admission as a practical primitive.

## Acknowledgements

This work was supported by the National Natural Science Foundation of China (NSFC) (Grant Nos. 62232005 and 62202126).

## Impact Statement

This paper studies a systems mechanism for improving the reliability and efficiency of serving multi-adapter (e.g., LoRA) workloads by enforcing a feasibility constraint at ingress, reducing tail-latency amplification due to adapter residency thrashing. The intended positive impact is to make model serving more predictable and resource-efficient,

which may reduce operational cost and energy consumption for deployed ML services.

Potential negative impacts are indirect. First, improved serving efficiency can lower the barrier to deploying large-scale inference, which could amplify downstream misuse of ML applications depending on the deployment context. Second, the mechanism supports prioritization (e.g., VIP versus background traffic) and could be used to reinforce unequal access if configured irresponsibly. Our contribution is a scheduling/serving control that does not change model capabilities or outputs; responsible use remains a deployment decision. We encourage practitioners to use such prioritization transparently, audit its effects, and apply appropriate governance and content-safety measures at the application level.

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

**Appendix roadmap.**

- Appendix A documents reproducibility and measurement details (setup, run protocol, logging, and metric computation).

- Appendix B reports the full baseline numbers used in the main figures/tables.

- Appendix C provides robustness controls that rule out load/throughput confounding.

- Appendix D expands the mechanistic stall diagnostic and supporting breakdowns.

- Appendix E documents CLIMB's implementation semantics and the full CLIMB pseudocode (corner cases, event model, and complexity).

- Appendix F records design-space explorations and policy variants for auditability (not promoted as mainline baselines).

- Appendix G replicates the cliff signature across LoRA ranks.

- Appendix H replicates the cliff signature and the GLOBALFIFO–CLIMB gap on RTX PRO 6000.

- Appendix I quantifies controller overhead.

- Appendix J provides extended related work and broader context (serving, PEFT, and scheduling).

## A. Reproducibility and Measurement Details

This appendix provides the implementation-facing details needed to reproduce our evaluation. In the main paper, we intentionally compress setup and instrumentation to preserve narrative focus; here we make the experimental assumptions *auditable*: what is fixed vs. swept, what is measured vs. derived, and how each reported metric is computed from logs. Throughout, we use the paper-facing policy names (e.g., CLIMB) and avoid internal identifiers.

**Scope and reader guide.** We organize reproducibility around four repeatedly referenced artifacts: (i) the concrete setup and run protocol (hardware/software/model/workload), (ii) the operational meaning of key symbols used in figures and tables, (iii) a mechanism matrix that makes policy differences checkable at a glance, and (iv) the minimum logging schema and aggregation rules that uniquely determine each metric. If you only need a "what to run" recipe, start from the setup checklists (Tables 2 and 3) and the artifact map below. If you want to validate metric semantics (e.g., why a p99 spike is queue-driven vs. engine-driven), read the measurement section and Table 8.

### A.1. Artifact Map and Run Protocol

**Artifact layout.** Our reproduction artifact[1] is intended to be a self-contained bundle. At minimum, it includes: (1) configuration files for each workload and policy, (2) a runner script that launches vLLM with the intended arguments and drives open-loop arrivals, (3) a parser that converts raw logs into per-run summaries, and (4) the plotting scripts used to generate the paper figures/tables. In addition, we include a dependency snapshot (e.g., `pip freeze`) and a model fingerprint (see below) to make the environment and weights immutable.

**Run unit and replication.** A single *run* is defined by a tuple

$$(\text{policy, workload, } K, \text{ model/LoRA settings, seed}),$$

executed for a fixed warmup duration followed by a fixed measurement duration. Unless explicitly noted (e.g., exploratory design-space probes), we run three seeds and report mean±std across seeds. Within each run, percentiles (p50/p90/p99) are computed *after warmup* over request-level samples, then aggregated across seeds. This "percentile-per-run, then average" convention avoids mixing distributions across independent runs.

---

[1] https://github.com/maxsignalll/CLIMB_artifact.

**What varies vs. what is fixed.** Most figures fix the model, LoRA rank, and serving stack, while varying one of: (i) the feasibility regime via $K$ (or a $K$ sweep), (ii) the working set size $W$ (via the number of distinct adapters), or (iii) the offered load (arrival rates or per-adapter rate). To keep the paper concise, we do not repeat these knobs in every appendix paragraph; instead, figure/table captions state the active sweep dimensions. The setup checklists record what is globally fixed (software stack, base model, LoRA rank, token limits) and what is parameterized (e.g., $K$).

### A.2. Setup Checklist

For readability, we split the setup checklist into environment/model settings (Table 2) and serving/workload settings (Table 3); this split is organizational and does not change the run protocol.

*Table 2.* **Reproducibility setup checklist: environment and model.** This table records hardware, software, model, and LoRA residency settings.

| Item | Value / Notes |
|---|---|
| *Hardware* | |
| GPU | RTX 5090 32GB (single GPU) |
| CPU / RAM | 25 vCPU, Intel Xeon Platinum 8470Q, 90GB RAM |
| OS | Ubuntu 22.04.1 LTS |
| NVIDIA driver / CUDA | Driver 580.95.05; CUDA 13.0 |
| *Software stack* | |
| Python | 3.10.19 |
| PyTorch (Paszke et al., 2019) | 2.9.0+cu128 |
| Transformers (Wolf et al., 2020) / Tokenizers | 4.57.3 / 0.22.2 |
| Serving engine | vLLM 0.13.0 (Kwon et al., 2023) (pip; full dependency list in the reproduction artifact) |
| Local patches | None |
| *Model & LoRA* | |
| Base model | Qwen2.5-7B-Instruct (Hui et al., 2024) |
| Model snapshot (weights SHA256) | `6252cd7a9d5d300781a5a0df1433e9a2f86848f4f19e8720` `f27f4e77200c8e3f` |
| Model snapshot (metadata SHA256) | `1460c7b5de33f0a828d92d6394a8a38270c6dd9cc6573172` `6ee991391a138617` |
| Precision | bfloat16 |
| LoRA rank | 128 |
| Adapter-residency capacity | Qwen2.5-7B rank-128 adapter: approx. 0.301 GiB; fully populated $K{=}4$: approx. 1.20 GiB; fully populated $K{=}8$: approx. 2.41 GiB |
| LoRA residency budget $K$ | Variable (default $K{=}4$ unless stated; sweeps as specified per figure/table caption) |
| Adapter library | 8 adapters total (1 VIP + 7 BG; mainline runs; W-sweep uses a subset per M) |

**Environment and engine configuration.** Table 2 summarizes the concrete hardware/software environment and the base model and LoRA settings. Table 3 records the vLLM knobs, initialization-reported KV-cache capacity, and workload protocol. We list explicit vLLM arguments because they can affect batching, memory pressure, and therefore whether a run enters the cliff regime. Any vLLM parameter not listed should be interpreted as "vLLM 0.13.0 default" (not silently tuned per experiment). We intentionally avoid disclosing machine-specific filesystem paths or cloud vendor details; only the reproducibility-relevant identifiers are recorded.

**Model fingerprinting (local snapshot).** Our runs load the base model from a local directory on the server. In this case, a HuggingFace "revision" string may not exist; instead, we recommend recording an immutable fingerprint of the local snapshot. Concretely, we compute a `weights.manifest` file as the sorted list of SHA256 hashes of weight shards (e.g., `*.safetensors`), and report the SHA256 of this manifest as the model-weight fingerprint. Optionally, we similarly fingerprint tokenizer/config files with a `meta.manifest` (see Table 2). This fingerprinting is sufficient for reproducibility even when the original download source was a domestic mirror.

**Workload protocol and invariants.** Our mainline experiments use open-loop (fixed-rate) arrivals so the offered load is controlled independent of system response (Jain, 1990). We fix prompt and generation lengths per class and keep the warmup/duration constant across comparable runs. When we perform $K$ or $W$ sweeps, we keep the arrival process and

*Table 3.* **Reproducibility setup checklist: serving and workload.** Explicitly configured knobs are listed; rows marked as defaults indicate vLLM 0.13.0 default settings.

| Item | Value / Notes |
|---|---|
| *Serving / engine knobs (vLLM)* | |
| Explicit vLLM args | `gpu_memory_utilization=0.85,` `max_model_len=4096,` `max_lora_rank=128,` `max_loras=K` |
| vLLM initialization report | Under `gpu_memory_utilization=0.85` and `max_model_len=4096`, vLLM reports 8.51 GiB available for KV cache and an aggregate GPU KV-cache capacity of 159,360 token slots at initialization; CLIMB does not use these values at runtime. |
| Batching args | Defaults in vLLM 0.13.0 (not explicitly set) |
| Swap / offload | Defaults in vLLM 0.13.0 (not explicitly set) |
| Runtime LoRA update | Enabled |
| Client harness | Open-loop load generator; inflight cap = 128 |
| *Workload generation & run protocol* | |
| Arrival mode | Open-loop (fixed-rate) |
| VIP parameters | prompt_len=64; rps=1; concurrency=1; 1 VIP adapter |
| BG parameters | prompt_len=2048; rps_each=3; concurrency_each=1; 7 BG adapters (mainline) |
| Generation length | max_tokens=64 |
| Warmup / duration | 60 / 600 seconds |
| Seeds | 101, 102, 103 (default; exploratory runs may use a single seed, explicitly noted) |
| Aggregation | Percentiles computed per run after warmup; mean±std across seeds |

token limits unchanged unless the caption states otherwise. If an exploratory run uses a single seed (for rapid iteration), we explicitly mark it as exploratory and do not mix it into the mainline aggregated tables.

**Workload families (W2 vs. W1).** Our evaluation uses open-loop (fixed-rate) arrivals (Table 4). The mainline results are driven by a phase-shift workload family, `W2_phase`, designed to make the residency cliff visible within a single run. Each `W2_phase` run consists of three 200 s phases (warmup excluded): (i) a low-load phase with one background adapter, (ii) an overload phase with a larger set of background adapters (peak working set), and (iii) a recovery phase returning to one background adapter. Across `W2_phase` sweeps we vary the *peak* working-set size by changing how many BG adapters receive arrivals (the overload-phase $W$), while keeping per-request prompt/generation lengths and the open-loop arrival process fixed. We use two `W2_phase` variants as targeted controls: a load-matched control (fewer BG adapters but higher per-adapter rate) and a reduced-load liveness probe to avoid saturation. Finally, we include an auxiliary family `W1_*` (e.g., `W1_main`, `W1_hotcold`) for safe-regime and skewed-demand checks; their exact parameters are specified in the workload config files.

*Table 4.* **Canonical `W2_phase` workloads.** All `W2_phase` runs use the same three-phase schedule (low-load $\rightarrow$ overload $\rightarrow$ recovery); $W_{\text{peak}}$ denotes the overload-phase working-set size. Other fixed parameters (VIP/BG prompt lengths, VIP rate, max_tokens, warmup/duration) are in Table 3.

| Workload variant | Purpose | $W_{\text{peak}}$ | BG rps_each |
|---|---|---|---|
| `W2_phase` (mainline; $M \in \{5, 6, 7, 8\}$) | Mainline $W/K$ sweeps | 5–8 | 3 |
| `W2_phase` (load-matched; $M = 4$) | Load-matched control | 4 | 7 |
| `W2_phase` (liveness; $M = 8$) | Reduced-load liveness probe | 8 | 1 |

### A.3. Symbols and Operational Definitions

**Notation cross-reference.** Table 5 lists the symbols that appear repeatedly in our analysis (e.g., $K$, $W_t$, $R_t$, $A_t$ and TTFT decomposition timestamps). This table is meant as a lookup so figures remain readable without reintroducing notation each time.

**Operational meaning in implementation.** Some symbols are defined mathematically in the main paper but must be *operationalized* to be reproducible. We clarify the key ones here. $R_t$ (resident set proxy) denotes the adapters considered

*Table 5.* **Evaluation symbols.** This table collects the notation repeatedly used in figures/tables.

| Symbol | Meaning (unit) |
|---|---|
| $K$ | LoRA residency budget (max #adapters simultaneously resident/ready; vLLM `max_loras`). |
| $R_t$ | Controller-reported resident-set proxy at time $t$ (from the controller log), with $|R_t| \leq K$. |
| $A_t$ | Controller admitted/active adapter set at time $t$ (eligible for dispatch). |
| $I_t$ | Set of adapters with requests currently in-flight (dispatched but not completed) in the engine at time $t$. |
| $W_t$ | Working set at time $t$ (set of distinct adapters with pending or in-flight demand); $|W_t|$ is its size. |
| $\lambda_{\mathsf{VIP}}, \lambda_{\mathsf{BG}}$ | Arrival rates of VIP/BG traffic (requests/s). |
| $\lambda$ | Window-level total arrival rate (VIP+BG), with $\lambda = \lambda_{\mathsf{VIP}} + \lambda_{\mathsf{BG}}$. |
| $r$ | A request; $a(r)$ is its adapter id; $c(r) \in \{\mathsf{VIP}, \mathsf{BG}\}$ is its class. |
| $t_{\mathrm{arr}}(r)$ | Request arrival timestamp (s). |
| $t_{\mathrm{disp}}(r)$ | Dispatch-into-engine timestamp (s). |
| $t_{\mathrm{ft}}(r)$ | First-token produced timestamp (s). |
| $\mathrm{queue}(r)$ | Queueing delay before dispatch: $t_{\mathrm{disp}}(r) - t_{\mathrm{arr}}(r)$ (s). |
| $\mathrm{engine}(r)$ | Engine delay after dispatch: $t_{\mathrm{ft}}(r) - t_{\mathrm{disp}}(r)$ (s). |
| $\mathrm{TTFT}(r)$ | Time-to-first-token: $\mathrm{queue}(r) + \mathrm{engine}(r)$ (s). |

resident by the controller at time $t$; we derive it from the controller log rather than engine snapshots. $A_t$ (controller admitted/active set) denotes the set of adapters eligible for dispatch at time $t$; it is derived from the controller's active-set fields. $W_t$ (working set) is the set of distinct adapters with pending or in-flight demand at time $t$; concretely, we define an adapter as "in $W_t$" if it has a non-empty queue or at least one in-flight request at $t$. When comparing policies, it is $|W_t|$ relative to $K$ that determines whether the run is in a safe regime ($|W_t| \leq K$) or a cliff regime ($|W_t| > K$) (Denning, 1968).

### A.4. Policy Feature Matrix (Mechanism Axes)

**Why this matrix exists.** Many baseline policies differ along multiple axes (admission/activation, dispatch, and optional guardrails). To prevent "paper-code mismatch" and to make comparisons auditable, Table 6 records these axes explicitly. This table is *not* a performance table; it is a mechanism checklist used to interpret why two policies behave differently under the same workload.

**Axis definitions and interpretation.** We distinguish **activation/admission** from **dispatch**. Activation/admission decides which adapters are allowed to occupy the limited residency budget $K$ (or be considered active); dispatch selects the next request among currently eligible adapters/queues. The **Gate** column indicates whether the policy enforces an explicit feasibility constraint to keep the admitted/active set within $K$. For example, CLIMB uses hard-gated admission: if the VIP working set saturates capacity, BG admission is paused so VIP is not blocked by BG occupancy. The **Guards** column indicates stability-oriented guardrails (e.g., lease time, switch budget, cooldown) that bound thrashing; these are separate from fairness mechanisms such as DRR deficit accounting (Shreedhar & Varghese, 1996). The **Locality** column marks policies that explicitly optimize locality (e.g., clustered queues) beyond feasibility control.

**Consistency with names.** The policy names used throughout the paper are chosen to match these axes. For instance, URGENCYGATE differs from CLIMB only in the activation choice for BG (urgency vs. RR), while keeping FIFO dispatch; CLASSDRR keeps the same admission skeleton as CLIMB but changes dispatch to class-level DRR with a fallback mode. This naming-by-mechanism ensures that a reader can predict, at a high level, what a policy changes without needing internal identifiers.

### A.5. Metrics, Aggregation, and Logging Schema

**Request-level timestamps and TTFT decomposition.** All latency metrics are computed from per-request timestamps recorded by the client harness and the serving engine. For a request $r$, we record arrival time $t_{\mathrm{arr}}(r)$, dispatch-into-engine time $t_{\mathrm{disp}}(r)$, and first-token time $t_{\mathrm{ft}}(r)$. We define queueing delay as $\mathrm{queue}(r) = t_{\mathrm{disp}}(r) - t_{\mathrm{arr}}(r)$ and engine delay as $\mathrm{engine}(r) = t_{\mathrm{ft}}(r) - t_{\mathrm{disp}}(r)$, so that $\mathrm{TTFT}(r) = \mathrm{queue}(r) + \mathrm{engine}(r)$. This decomposition is essential in the cliff regime: policies can have similar overall TTFT while failing for different reasons (e.g., long pre-dispatch queueing vs. long post-dispatch stalls). Unless stated otherwise, we report p99 over the post-warmup interval, computed separately for VIP and BG request subsets (Agrawal et al., 2024).

*Table 6.* **Policy feature matrix. Gate**: enforces the ingress feasibility contract (keeps $|A_t \cup I_t| \leq K$). **Guards**: stability guardrails (lease/budget). **Local**: locality-aware dispatch. This matrix highlights mechanism differences across the design space.

| Policy | Gate | Activation | Dispatch | Guards | Local | Notes |
|---|---|---|---|---|---|---|
| CLIMB | ✓ | RR activation (class-aware) | FIFO | ✗ | ✗ | BG admission pauses when VIP backlog exceeds $K$. |
| GATEDRR-GUARD | ✓ | DRR fairness | DRR | ✓ | ✗ | Guardrails: lease / switch budget / cooldown. |
| OPENDRR-GUARD | ✗ | DRR fairness | DRR | ✓ | ✗ | No hard gate (stress-test gating necessity). |
| GATERR-GUARD | ✓ | RR | RR | ✓ | ✗ | RR-style scheduling; no deficit accounting. |
| GATEDRR | ✓ | DRR fairness | DRR | ✗ | ✗ | No stability guardrails. |
| CLASSDRR | ✓ | Same as CLIMB | Class-level DRR / FIFO fallback | ✗ | ✗ | BINDING: class-DRR + class-RR; SLACK: fallback. |
| URGENCYGATE | ✓ | Urgency-based | FIFO | ✗ | ✗ | Urgency: max HOL age, $\alpha \cdot$ queue length. |
| BGCAP | ✓ | VIP pin + BG distinct-cap | FIFO | ✗ | ✗ | BG active set capped by `bg_cap`. |
| LRUGATE | ✓ | LRU eviction on misses | cluster_q | ✗ | ✓ | Locality-aware dispatch via clustered queues. |
| LOCKGATE | ✓ | Lock resident set until drain | FIFO | ✗ | ✗ | Suppresses switching by locking active set once filled. |
| GLOBALFIFO | ✗ | – | global FIFO | ✗ | ✗ | No admission control; single global FIFO queue. |

*Table 7.* **Per-class throughput and token-equivalent throughput** (`W2_phase`, M8; mean $\pm$ std over 3 seeds). Per-class rps is computed from completed requests over the post-warmup measurement window used for latency metrics (warmup excluded). TokEq/s is an *equivalent* throughput based on configured token budgets: TokEq/s $= \text{Thr}_{\text{VIP}} \cdot (L_{\text{vip}} + T) + \text{Thr}_{\text{BG}} \cdot (L_{\text{bg}} + T)$, with $L_{\text{vip}}{=}64$, $L_{\text{bg}}{=}2048$, and $T{=}64$; it is not the realized token rate.

| $K$ | Policy | VIP rps | BG rps | Total rps | TokEq/s |
|---|---|---|---|---|---|
| 4 | GLOBALFIFO | $0.955 \pm 0.049$ | $9.700 \pm 0.216$ | $10.655 \pm 0.226$ | $20609.3 \pm 456.4$ |
| | CLIMB | $0.955 \pm 0.049$ | $9.701 \pm 0.215$ | $10.656 \pm 0.225$ | $20610.2 \pm 454.0$ |
| 8 | GLOBALFIFO | $0.955 \pm 0.049$ | $9.705 \pm 0.210$ | $10.660 \pm 0.220$ | $20619.9 \pm 444.4$ |
| | CLIMB | $0.955 \pm 0.049$ | $9.700 \pm 0.215$ | $10.655 \pm 0.225$ | $20609.7 \pm 455.5$ |

**Zero-failure criterion.** We verify that all runs eventually complete all generated requests (no timeouts, cancellations, or engine errors). Achieved throughput can still differ across policies (e.g., due to admission throttling or controller-induced idling), so we explicitly report throughput to expose throughput confounding and avoid interpreting load-shedding as latency improvements. For the main GLOBALFIFO vs. CLIMB comparisons in sweeps, we additionally confirm throughput is matched within noise.

**Throughput and token-equivalent throughput.** Achieved throughput (rps) is computed from *completed* requests over the same post-warmup measurement window used for latency percentiles (Table 7), reported per-class (VIP, BG) and in total. Because request costs differ substantially across classes in our workload (e.g., prompt lengths), we additionally compute a token-equivalent throughput metric (TokEq/s) from configured token budgets: TokEq/s $= \text{Thr}_{\text{VIP}} \cdot (L_{\text{vip}} + T) + \text{Thr}_{\text{BG}} \cdot (L_{\text{bg}} + T)$, with $L_{\text{vip}}{=}64$, $L_{\text{bg}}{=}2048$, and $T{=}64$. TokEq/s is an *equivalent* throughput for auditing throughput matching; it is not the realized token rate.

**Starvation / liveness metrics.** When we report BG liveness, we use a backlogged-based definition that only counts intervals where BG demand exists. Intuitively, a BG adapter is "backlogged" when it has pending requests but receives no service; liveness gaps capture the longest such interval. We use `backlogged_wait: t_served - t_start`, where `t_start` is the first arrival when backlog goes $0{\rightarrow}{>}0$ and `t_served` is the first dispatch. If liveness is not reported for a particular figure/table, we do not infer it from unrelated counters.

**Engine events and residency-related counters.** Some analyses refer to residency dynamics (e.g., swapping, loading/eviction, or active-set churn). These quantities can be logged either as event traces (each load/evict with a timestamp and adapter id) or as periodic snapshots (sample $|R_t|$ and $|A_t|$). For reproducibility, we record the sampling period or event

*Table 8.* **Minimum logging schema.** Timestamps use a monotonic clock (seconds since run start); listed fields suffice to reproduce all metrics in this paper.

| Quantity | Log source | Field / computation |
|---|---|---|
| $t_{\mathrm{disp}}(r)$ | `requests_log.csv` | `dispatch_ts`; queue$(r) = t_{\mathrm{disp}} - t_{\mathrm{arr}}$ |
| $t_{\mathrm{ft}}(r)$ | `requests_log.csv` | `first_token_ts`; engine$(r) = t_{\mathrm{ft}} - t_{\mathrm{disp}}$; TTFT$(r) = t_{\mathrm{ft}} - t_{\mathrm{arr}}$ |
| Completion time $t_{\mathrm{end}}(r)$ (used for throughput) | `requests_log.csv` | `finish_ts`; throughput from completions per unit time (TokEq/s uses fixed token counts) |
| $\lvert A_t \rvert$ (controller admitted set size) | `control_log.csv` | $\lvert A_t \rvert = \lvert$`active_vip`$\rvert + \lvert$`active_bg`$\rvert$ (list lengths) |
| Load/evict / swap events (if used) | `control_log.csv` | `swap_event_count`; switching proxy via $\Delta$`swap_event_count`$/\Delta t$ (+`swap_bytes_total`, `swap_ms_total`) |

semantics used. We use controller-side proxy counters in controller log, not engine-side events; we do not use periodic engine snapshots. If a quantity is not logged in the artifact, it is not used to support any main claim.

**Aggregation and warmup handling.** All metrics are computed after discarding the warmup interval. For each run, we compute percentiles on the filtered samples; we then aggregate across seeds as mean±std of these per-run statistics. When a run produces multiple log files (client-side arrival/completion records and engine-side timing records), we join them by a stable request identifier. This join rule is critical for separating VIP vs. BG distributions and for attributing each request to an adapter.

**Minimum logging schema.** Table 8 lists the minimal fields needed to reproduce every reported metric. We intentionally keep it small: request id, class, adapter id, the three timestamps for TTFT decomposition, and (optionally) residency counters and event traces. If you extend the analysis with additional metrics, we recommend adding rows to this schema rather than ad-hoc logging, so that every derived quantity has a documented provenance.

# B. Detailed Baseline Performance and Trade-Offs

This appendix reports the comprehensive performance characterization of all baselines that are referenced throughout the evaluation, including the baseline trade-off plot and several discussion points about throughput confounding. The goal of this section is deliberately simple: when a reader asks *"what happens if we do nothing?", "what if we use a locality heuristic?", or "what is the cost of pinning VIP?"*, they should be able to answer it by inspecting a single table with clear semantics.

**Why we report two regimes (Cliff vs. Safe Anchor).** We evaluate each baseline at a fixed working set size $W=8$ under two residency budgets: (i) a **cliff regime** with $K=4$, where $W > K$ and the engine is expected to encounter non-resident adapters on the critical path, and (ii) a **safe-anchor regime** with $K=8$, where $W \approx K$ and residency stalls should largely disappear. Reporting both regimes serves two purposes. First, it separates "how a policy behaves when feasibility is violated" from "how intrusive a policy is when feasibility holds". Second, it guards against over-interpreting a single operating point: a policy that looks competitive in the cliff regime may simply be throttling throughput, while a policy that is excellent at the safe anchor but collapses in the cliff regime may be brittle to modest demand shifts.

**How to read the table.** Table 9 reports VIP TTFT p99 and decomposes it into **queue** vs. **engine** components, along with BG TTFT p99 and overall throughput (rps). The decomposition is essential for interpreting the residency cliff. A large *engine* component indicates that requests are stalling after dispatch (consistent with residency faults and engine-side stalls), while a large *queue* component indicates the delay is primarily outside the engine (e.g., admission/activation bottlenecks or FIFO waiting). Throughput is reported alongside latency because some baselines can achieve small p99 only by serving fewer requests per unit time; such points should not be interpreted as "strictly better" in the usual latency–throughput sense. All values are mean±std over three seeds, computed per run after warmup.

**Key observations in the cliff regime** ($K=4$). GLOBALFIFO represents the "do nothing" baseline and exhibits a clear cliff signature: VIP p99 is dominated by the *engine* component, indicating engine-side stalls rather than pure queueing.

*Table 9.* **Detailed Baseline Performance** ($W$=8). Detailed comparison of all baselines under Cliff ($K$=4) and Safe ($K$=8) regimes. BGCAP suffers from high BG latency; LOCKGATE suffers from low throughput. CLIMB provides a strong throughput-preserving trade-off. Values are mean±std (3 seeds).

| | VIP Latency p99 (s) | | | BG p99 | Thr |
|---|---|---|---|---|---|
| Policy | Total (TTFT) | Queue | Engine | (s) | (rps) |
| *Regime: Cliff* ($K$=4) | | | | | |
| GLOBALFIFO | 38.71±13.2 | 2.10±0.1 | 38.43±12.9 | 20.17±1.9 | 10.66±0.2 |
| LRUGATE | 40.96±10.4 | 40.85±10.4 | 0.27±0.2 | 31.21±7.0 | 10.66±0.2 |
| BGCAP | 3.77±0.6 | 3.51±0.8 | 0.19±0.1 | 118.40±0.2 | 6.89±0.8 |
| LOCKGATE | 0.05±0.0 | 0.00±0.0 | 0.05±0.0 | 0.06±0.0 | 3.99±0.2 |
| CLIMB | 13.69±3.4 | 13.60±3.4 | 0.13±0.0 | 40.36±5.0 | 10.66±0.2 |
| *Regime: Safe Anchor* ($K$=8) | | | | | |
| GLOBALFIFO | 0.13±0.0 | 0.00±0.0 | 0.13±0.0 | 0.15±0.0 | 10.66±0.2 |
| LRUGATE | 0.13±0.0 | 0.00±0.0 | 0.13±0.0 | 0.15±0.0 | 10.66±0.2 |
| BGCAP | 3.85±0.5 | 3.64±0.4 | 0.21±0.1 | 118.30±0.4 | 6.78±0.6 |
| LOCKGATE | 0.05±0.0 | 0.00±0.0 | 0.05±0.0 | 0.06±0.0 | 3.99±0.2 |
| CLIMB | 0.13±0.0 | 0.00±0.0 | 0.13±0.0 | 0.14±0.0 | 10.66±0.2 |

LRUGATE (locality-driven) shifts the dominant contribution to the *queue* component in this workload, illustrating an important point: locality heuristics can change *where* delay accumulates without necessarily restoring feasibility. BGCAP is an extreme "protect VIP" knob (pin VIP and cap BG), which can make VIP p99 small but at the cost of very large BG tail latency and reduced throughput—a reminder that aggressive protection can implicitly act as throttling or starvation for BG. LOCKGATE can produce very small tail latencies for both classes in this operating point, but it also significantly reduces throughput; we treat it as a *throughput-confounded* extreme and do not use it to claim Pareto improvements. Finally, CLIMB is designed to remove engine-side residency stalls for VIP while preserving throughput: in the cliff regime it keeps the VIP *engine* component near the safe-anchor level and externalizes waiting into the *queue* component, with a moderate BG tail increase. This is the intended trade-off pattern for feasibility restoration under fixed budget $K$.

**Key observations at the safe anchor** ($K$=8). At $K$=8, the policies that do not aggressively throttle (GLOBALFIFO, LRUGATE, CLIMB) converge to similar VIP and BG tails, consistent with the expectation that the residency cliff is avoided when $W \leq K$. This regime is primarily a **transparency check**: a feasibility-restoring mechanism should not impose unnecessary overhead when the system is already in a feasible/fast regime. In contrast, policies that encode hard protection or hard locking (BGCAP, LOCKGATE) retain their characteristic signatures even at the anchor (e.g., reduced throughput or extreme behavior), which is why we treat them as "stress points" rather than general-purpose solutions.

**Caveats and intended use.** This baseline comparison is a point evaluation at a fixed $W$=8 and two representative $K$ values. It is not meant to replace the main sweep results (which establish the knee law and the mechanism closure), but to make baseline trade-offs concrete and to prevent ambiguity in later discussions (e.g., "is this improvement due to throttling?", "did the baseline move delay from engine to queue?"). When interpreting the table, readers should focus on (i) the TTFT decomposition to identify the dominant failure mode, and (ii) throughput to rule out trivial gains due to under-service. We use this table as the numeric backing for the baseline trade-off visualization and as a reference for baseline selection in the main text.

## C. Robustness Controls

This appendix provides two targeted controls that rule out common alternative explanations for the observed cliff behavior. In cliff-driven systems, it is easy to draw incorrect conclusions from (i) *load confounding*—changing the working set often changes the offered load, and (ii) *throughput confounding*—a policy can appear to improve tail latency simply by throttling service. Our main results emphasize that the LoRA residency cliff is *boundary-driven* (triggered by $W > K$) and *stall-driven* (dominated by engine-side delay). The controls below stress-test these claims under minimal perturbations.

**What is being controlled.** Control **D** addresses the fact that in a working-set sweep, increasing $W$ typically increases offered BG load (more distinct BG adapters, each with its own request stream). A naive explanation for a tail blow-up is therefore "we simply ran a heavier workload." Control **H** addresses a different concern: a $K$-sweep changes feasibility

*Table 10.* **Controls (D/H).** VIP TTFT p99 and its queue/engine components plus throughput. Values are mean±std over seeds 101/102/103. Units are seconds (s) except throughput (rps).

| Control | Setting | VIP TTFT p99 | VIP q p99 | VIP eng p99 | Thr |
|---|---|---|---|---|---|
| D (load-matched) | $M8, K{=}4$ (ref.) | 38.71±13.18 | 2.10±0.08 | 38.43±12.91 | 10.66±0.18 |
| D (load-matched) | $M4, K{=}2$ (matched) | 53.04±18.80 | 3.59±0.18 | 53.04±18.80 | 13.13±0.26 |
| H (same arrivals) | $K{=}8$ | 0.13±0.00 | 0.00±0.00 | 0.13±0.00 | 10.66±0.22 |
| H (same arrivals) | $K{=}7$ | 45.22±13.82 | 0.01±0.00 | 45.22±13.82 | 10.66±0.22 |

and can also change throughput; a knee could be misread as a throughput artifact. We therefore include a one-slot change ($K{=}8 \rightarrow 7$) under the *same* arrival rates to isolate the discrete boundary effect.

### C.1. Control D: Load-Matched Working-Set Scaling

**Design rationale (Control D: matched-load pairing).** In the $W$-sweep (Section 5.1), $W$ is increased by adding more BG adapters. Because each BG adapter is driven by a fixed-rate stream in our open-loop harness, this increases the aggregate offered BG load. Control D constructs a paired setting where total offered load is comparable while feasibility pressure remains qualitatively the same: we scale the number of distinct adapters and the residency budget together, for example comparing ($W{=}8, K{=}4$) with ($W{=}4, K{=}2$), and adjust the BG stream rates so that the aggregate offered load is comparable across the pair (we use `BG rps_each=3` for ($W{=}8, K{=}4$) with 7 BG adapters, and `BG rps_each=7` for ($W{=}4, K{=}2$) with 3 BG adapters; `VIP rps=1` in both, yielding matched total BG load of 21 rps). Intuitively, this control asks: *if we remove "more BG load" as an explanation, does the cliff signature still appear whenever $W > K$?*

**What to look for.** A load-confounded cliff would typically manifest as (i) a large increase in *queueing* delay (waiting before dispatch) and/or (ii) a throughput drop consistent with classical saturation. In contrast, a stall-driven residency cliff should show (i) tail latency dominated by the *engine* component and (ii) throughput that remains broadly comparable (or at least does not collapse proportionally to p99). In Table 10, Control D exhibits the latter pattern: despite comparable offered load and a scaled-down $(W, K)$ pair, VIP TTFT p99 remains extremely large and is dominated by the engine component, while throughput does not exhibit a corresponding collapse. This supports the interpretation that the blow-up is not simply "more load," but the nonlinear amplification caused by fetch-induced stalls once feasibility is violated.

**Interpretation and limits.** Control D is intentionally coarse: it does not attempt to match every micro-level condition (e.g., exact cache hit patterns), but to remove the most obvious confound—aggregate offered load—while preserving the feasibility violation ($W > K$). The fact that the tail remains engine-dominated under the matched pair is consistent with our mechanism claim: when the working set exceeds the discrete residency budget, rare misses can enter the engine critical path, stall continuous batching, and amplify tails without requiring a classical throughput collapse. We treat this as robustness evidence rather than a primary result, and we do not overfit conclusions to the absolute p99 magnitudes, which can vary with the exact adapter library and traffic mix.

### C.2. Control H: Same Arrival Rates, One-Slot Budget Change

**Design rationale.** Control H is a near-minimal perturbation that isolates the discrete boundary effect. We fix the working set ($W$), fix the open-loop arrival process ($\lambda_{\text{VIP}}, \lambda_{\text{BG}}$), and change only the residency budget by one slot: $K{=}8$ (safe anchor) vs. $K{=}7$ (one-slot reduction). This construction keeps throughput essentially matched by design (Table 10), so any dramatic change in tail behavior cannot be attributed to throttling or a different offered load. It also directly targets the core hypothesis: the cliff is triggered when the system crosses a *discrete* feasibility boundary.

**What to look for.** Under a boundary-driven cliff, we expect a qualitative regime switch: at $K{=}8$ (with $W \leq K$), TTFT tails should remain in the fast regime; at $K{=}7$ (with $W > K$), tails should jump sharply even though throughput is essentially unchanged. Moreover, a stall-driven mechanism predicts that this jump should appear primarily in the *engine* component, because the additional cost is incurred after dispatch when a non-resident adapter surfaces on the engine critical path. Table 10 shows exactly this pattern: the safe anchor has small VIP TTFT p99, while a one-slot reduction causes an abrupt jump to a cliff-scale p99, with queue p99 remaining negligible and the engine component accounting for essentially the entire tail. This is strong evidence that the transition is boundary-driven rather than a smooth saturation effect.

**Interpretation and limits.** Control H should be read as a "one-bit" feasibility test: a tiny change in $K$ flips the system from feasible to infeasible, and the tail response is correspondingly discontinuous. The key message is not the specific p99 value at $K=7$, but the qualitative regime switch under matched throughput. This is consistent with readiness atomicity: once $W$ exceeds $K$, some requests inevitably encounter a missing adapter, and when such misses occur on a coupled engine cadence, they induce system-wide amplification beyond the culprit request. We use this control to justify focusing on feasibility-first admission (as in CLIMB), which prevents these misses from entering the engine critical path in the first place.

### C.3. Summary: What the Controls Rule Out

Taken together, Controls D and H rule out two simple alternative explanations for the cliff: (1) that the tail blow-up is merely caused by increasing offered BG load as $W$ increases, and (2) that the knee observed in a $K$ sweep is an artifact of throughput changes. Instead, both controls support the main narrative: the residency cliff is a *discrete, feasibility-triggered* phenomenon ($W > K$) whose tail is dominated by *engine-side stalls*, motivating controllers that explicitly enforce feasibility and move unavoidable waiting into managed queueing at ingress.

## D. Mechanistic Stall Diagnostic: Expanded Analysis

This appendix expands on the mechanism-driven diagnostic introduced in Eq. (2). Our goal is not to predict run-level p99 (which is often unstable under phase-mixed workloads), but to provide an interpretable *window-level* indicator of when the system is operating near the stall-driven residency cliff. The diagnostic is *not* used by CLIMB; rather, it serves as additional evidence that the cliff is fundamentally an engine-side stall phenomenon triggered by discrete feasibility violations ($K < W$).

### D.1. Why Window-Level (and Why Not Run-Level p99)

Run-level p99 can be noisy in our setting for two reasons. First, cliff-inducing probes (e.g., phase-shift workloads) intentionally mix feasible and infeasible periods within a single run, so a single scalar p99 conflates multiple regimes. Second, continuous batching can amplify rare stalls into heavy-tailed delay distributions, making high quantiles sensitive to a small number of extreme events. To obtain a more stable signal, we operate on fixed-length windows and ask a simpler question: *is this window in a "bad" (stall-dominated) state or a "good" (fast-regime) state?*

### D.2. A Toy Abstraction with Formal Implications (Boundary and Knee)

We now make the toy abstraction behind Eq. (2) slightly more formal. The goal is *not* to model every detail of continuous batching, but to expose a small set of assumptions under which (i) a discrete feasibility violation ($K < W$) naturally induces a regime switch, and (ii) the resulting "knee" scales approximately linearly with the working set size.

**Assumption D.1** (Window-level mixing and coarse miss pressure)**.** Fix a short window and let $W_{\text{window}}$ be the number of distinct adapters appearing in that window (VIP+BG). We assume that when $K < W_{\text{window}}$, the engine cannot simultaneously keep all demanded adapters resident, so non-resident slow paths occur with non-negligible frequency. We summarize this effect by a scalar miss-pressure proxy $p_{\text{miss}}(W_{\text{window}}, K) \in [0, 1]$ that is monotone in $W_{\text{window}}/K$ and satisfies $p_{\text{miss}} \approx 0$ when $K \geq W_{\text{window}}$.

**Lemma D.2** (A symmetry-based miss proxy under uniform mixing)**.** *Under a uniform-mixing approximation where (i) requests in the window are equally likely to reference any of the $W_{\text{window}}$ active adapters and (ii) the engine can cover any $K$ of them at a time, the probability that a random request references a non-covered adapter is*

$$p_{\text{miss}} \approx \max\left(0, 1 - \frac{K}{W_{\text{window}}}\right).$$

*Proof sketch.* When $K \geq W_{\text{window}}$, all demanded adapters can be covered and the miss probability is $0$. Otherwise, by symmetry, for a uniformly sampled request the probability that its adapter lies in a size-$K$ covered subset of the $W_{\text{window}}$ active adapters equals $K/W_{\text{window}}$, hence miss probability $1 - K/W_{\text{window}}$. We use this only as a coarse proxy: real engines exhibit temporal correlation and non-uniform popularity, which is why we treat $p_{\text{miss}}$ as "miss pressure" rather than an exact cache-miss probability. $\square$

**Definition D.3** (Toy effective service time and effective load). Let $\lambda$ denote the total arrival rate (VIP+BG) in the window and let $s_0$ be the baseline fast-regime VIP engine time measured at a safe anchor ($K \geq W$). We model the VIP engine time per request as

$$S \approx s_0 + \tau \cdot X, \quad X \sim \text{Bernoulli}(p_{\text{miss}}),$$

where $\tau$ is an *effective* stall time that absorbs both a non-resident slow path and its cadence-coupled amplification. This yields $\mathbb{E}[S] \approx s_0 + p_{\text{miss}}\tau$ and the effective-load score $\rho_{\text{eff}} \triangleq \lambda\mathbb{E}[S] = \lambda(s_0 + p_{\text{miss}}\tau)$ (Eq. (2)).

**Proposition D.4** (Tail sensitivity as effective load approaches saturation (toy)). *In standard single-bottleneck queueing models, high-percentile waiting time increases rapidly as the utilization $\rho = \lambda\mathbb{E}[S]$ approaches $1$. For example, in an M/G/1 queue (Pollaczek–Khinchine), the mean waiting time satisfies*

$$\mathbb{E}[W_q] = \frac{\lambda\mathbb{E}[S^2]}{2(1-\rho)},$$

*which diverges as $\rho \uparrow 1$ (Harchol-Balter, 2013). Thus, even if the precise serving engine does not satisfy M/G/1 assumptions, $\rho_{\text{eff}}$ provides a mechanism-consistent indicator: once miss pressure consumes the system's slack, tails can grow disproportionately and appear "cliff-like".*

*Proof sketch.* The displayed formula is the classical M/G/1 result; it shows the generic $1/(1-\rho)$ blow-up near saturation. We use this only as an interpretive lens: continuous batching introduces additional coupling, and our $\tau$ is an *effective* parameter capturing both direct stall and amplification. $\square$

**Corollary D.5** (Near-linear knee law under the miss proxy). *Assume $K < W_{\text{window}}$ and use the proxy $p_{\text{miss}} = \max(0, 1 - K/W_{\text{window}})$ (Lemma D.2). If we define the knee as the boundary where $\rho_{\text{eff}}$ crosses a constant level (e.g., $\rho_{\text{eff}} = 1$ in the toy queueing lens), then solving $\rho_{\text{eff}} = \lambda(s_0 + \tau(1 - K/W_{\text{window}}))$ for $K$ gives*

$$K^{\star}(W_{\text{window}}) \approx W_{\text{window}}\left(1 - \frac{1/\lambda - s_0}{\tau}\right), \quad \text{clipped to } [0, W_{\text{window}}].$$

*Hence the predicted knee $K^{\star}$ scales approximately linearly with $W_{\text{window}}$ for fixed $(\lambda, s_0, \tau)$.*

*Proof sketch.* Rearrange $\rho_{\text{eff}} = \lambda(s_0 + \tau(1 - K/W_{\text{window}}))$: $1/\lambda = s_0 + \tau - \tau K/W_{\text{window}} \Rightarrow K/W_{\text{window}} = 1 - (1/\lambda - s_0)/\tau$. The clipping reflects that $K^{\star}$ cannot exceed $W_{\text{window}}$ and is nonnegative. $\square$

**Corollary D.6** (Regime conditioning of the stall term (by construction)). *If $K \geq W_{\text{window}}$, then $p_{\text{miss}} = 0$ and $\rho_{\text{eff}} = \lambda s_0$, independent of $\tau$. If $K < W_{\text{window}}$, then $p_{\text{miss}} > 0$ and $\rho_{\text{eff}}$ increases linearly with $\tau$.*

*Proof.* Immediate from the definition $p_{\text{miss}} = \max(0, 1 - K/W_{\text{window}})$. $\square$

**Proposition D.7** (Feasibility-first admission shifts delay from engine stalls to explicit queueing). *Suppose a controller enforces that at any time the set of adapters eligible to dispatch has size at most $K$, and suppose the engine can keep these admitted adapters device-resident in steady state (up to cold starts). Then admitted traffic has $p_{\text{miss}} \approx 0$ and experiences fast-regime engine time $\approx s_0$; any additional demand must wait* before *dispatch, manifesting primarily as explicit ingress queueing rather than amplified engine-side stalls.*

*Proof sketch.* If the admitted set size is at most $K$, a residency configuration exists that covers all admitted adapters simultaneously. Under the steady-state assumption that the engine maintains this configuration, non-resident slow paths for admitted traffic vanish ($p_{\text{miss}} \approx 0$), so the engine component remains near $s_0$. Any demand beyond the budget is necessarily withheld at ingress, which by definition contributes to pre-dispatch queueing. $\square$

*Remark* D.8 (How to interpret $\tau$). The parameter $\tau$ is *not* a physical adapter load time. It is an effective coefficient that captures both the direct non-resident slow path and its amplification under cadence coupling. This is why we fit $\tau$ once as a global constant and evaluate separability (ROC/AUC) rather than predicting absolute p99 values.

### D.3. Operational Definitions and Estimation

We instantiate the diagnostic at a fixed window size ($10\,\text{s}$ in our analysis).

**Window construction and filtering.** We slice each run (after warmup) into contiguous, non-overlapping windows of length 10 s, aligned to the run start. Within each window, we compute (i) total arrivals (VIP+BG) to estimate $\lambda$, (ii) $W_{\text{window}}$ as the number of distinct adapters appearing in that window (VIP and BG combined), and (iii) the labeling statistic (VIP engine p95) using the engine-delay definition in Appendix A. Operationally, we estimate $W_{\text{window}}$ by counting distinct adapter ids with arrivals in the window. To avoid unstable quantiles, we exclude windows with fewer than 10 VIP requests. We also treat the final partial window as `kept if it meets the same minimum-VIP threshold; otherwise dropped`. All window-level quantities are computed from the same joined request records used for the main metrics; the diagnostic does not require additional "non-resident" event instrumentation.

**Baseline engine time $s_0$ (safe anchor).** We measure $s_0$ from a safe-anchor configuration where feasibility is not at risk ($K \geq W$) and treat it as a constant reference for the fast regime. Concretely, we use the VIP engine p50 at the safe anchor.

**Arrival rate $\lambda$.** For each window, $\lambda$ is the number of total (VIP+BG) requests arriving in that window divided by the window duration.

**Miss-pressure proxy $p_{\text{miss}}$.** Let $W_{\text{window}}$ be the number of *distinct* adapters that appear in the window (including VIP). We approximate miss pressure by the fraction of demanded adapters that cannot be simultaneously covered by a budget of $K$ slots:

$$p_{\text{miss}} \triangleq \max\left(0,\, 1 - \frac{K}{W_{\text{window}}}\right).$$

This proxy is intentionally simple: it is designed to capture the discrete feasibility boundary ($K < W$) without requiring low-level cache-instrumentation of non-resident events.

**Effective stall time $\tau$.** The parameter $\tau$ is an *effective* stall time that absorbs both the direct cost of a non-resident slow path and its amplification through cadence coupling. We choose $\tau$ by grid search (0–200 s, step 5 s) to maximize separability of bad versus good windows on a designated diagnostic dataset.

**Selecting $\tau$ without per-setting tuning.** Although $\tau$ could be fit separately per workload, we intentionally use a single global value to avoid overfitting. Concretely, we run the grid search once on a designated diagnostic dataset and reuse the same $\tau$ for all other analyses and controls (no refitting). Figure 5(a) shows that separability typically saturates over a range of $\tau$; we pick the smallest value near this plateau (default $\tau=5$ s) as a conservative setting.

### D.4. Bad-Window Labeling and Evaluation Protocol

We label a window as *bad* if a robust tail statistic of VIP engine delay exceeds a multiplicative threshold of the safe anchor:

$$\text{bad window} \iff \text{VIP engine p95} > \gamma \cdot s_0, \quad \gamma = 5.$$

We use engine p95 (rather than run-level p99) to reduce sensitivity to single outliers while still capturing the onset of stall-dominated behavior. Given a scalar score per window ($\rho_{\text{eff}}$), we evaluate separability with ROC/AUC (Mann–Whitney AUC) (Fawcett, 2006; Hanley & McNeil, 1982).

### D.5. Results: Regime-Conditioned Value of the Stall Term

Table 11 summarizes two validations. On the full K-sweep windows, incorporating the stall term ($\tau=5$) improves AUC from 0.969 to 0.998. On the same-$\lambda$ different-miss control ($K=8$ versus $K=7$), the stall term yields a larger gain (0.916 to 1.000), consistent with the intended causal structure: holding the arrival process (and thus $\lambda$) essentially fixed, the cliff is triggered purely by crossing the feasibility boundary ($K < W$), and the resulting degradation is engine-side. Figure 5 further reports the $\tau$-sensitivity curve and ROC comparisons.

**Why the stall term is regime-conditioned.** When $K \geq W$, $p_{\text{miss}} \approx 0$ and Eq. (2) automatically reduces to the baseline load proxy $\lambda s_0$. In this feasible regime, stall events are rare, and adding an explicit stall term cannot materially improve separability. In contrast, when $K < W$, miss pressure is non-negligible and the stall term captures the key mechanism: non-resident slow paths on the engine critical path can consume slack and trigger sharp tail amplification.

*Table 11.* **Window-level diagnostic separability (AUC).** Adding the stall term ($\tau > 0$) improves separability most strongly when $K < W$ (miss pressure is non-negligible).

| Setting | AUC ($\tau = 0$) | AUC ($\tau = 5$) | $\Delta$ |
|---|---|---|---|
| K-sweep, full windows (27 runs / 789 windows) | 0.969 | 0.998 | +0.029 |
| Same $\lambda$, different miss ($K = 8$ vs. $K = 7$) | 0.916 | 1.000 | +0.084 |

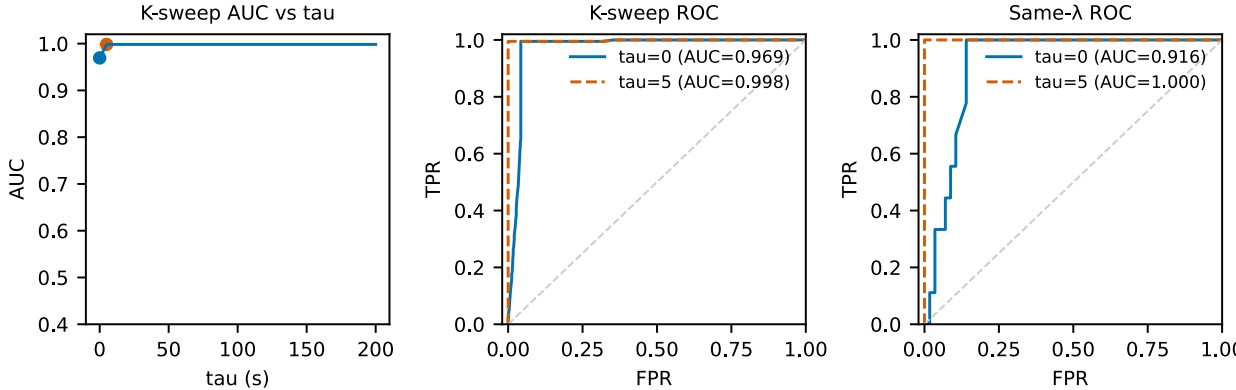

*Figure 5.* **Expanded diagnostic analysis.** (a) AUC versus $\tau$ on the K-sweep dataset (grid search 0–200s, step 5s). (b) ROC curves for $\tau=0$ vs. $\tau=5$ on the K-sweep dataset. (c) ROC curves for the same-$\lambda$ different-miss control ($K=8$ vs. $K=7$). Incorporating the stall term improves separability in the $K < W$ regime and vanishes automatically when $K \geq W$ (since $p_{\mathrm{miss}} \approx 0$).

### D.6. Limitations and How to Interpret the Diagnostic

This diagnostic is intentionally conservative in scope. It is a mechanism-consistent *risk score*, not a predictive model for exact tail percentiles. The fitted $\tau$ should not be interpreted as a physical "adapter load time"; it is an effective parameter that depends on engine configuration and the degree of cadence coupling. Finally, $p_{\mathrm{miss}}$ is a proxy based on working-set feasibility rather than direct instrumentation of non-resident events. Despite these limitations, the diagnostic is useful for triangulating mechanism: it explains why the cliff is boundary-triggered, why it is engine-side, and why feasibility-first controls such as CLIMB can mitigate it by keeping non-resident slow paths off the engine critical path.

**Sensitivity notes.** The diagnostic involves three user-facing knobs: the window length, the bad-window threshold $\gamma$, and the stall coefficient $\tau$. In this study we fix the window length to 10 s and $\gamma = 5$, and only sweep $\tau$ (0–200 s, step 5 s) to assess separability (Figure 5(a), Table 11). We do not report additional sensitivity sweeps over window length or $\gamma$; these can be added in future work if needed. Finally, this diagnostic is used only for analysis and is not part of CLIMB's control loop, so these knobs do not affect system behavior.

### D.7. VIP-Absence Intervals in Mild Overload ($W = K+1$)

To substantiate Remark 3.2, we measure *VIP-absence intervals* from controller logs(Table 12): intervals where VIP demand is backlogged yet there is no active/in-flight VIP work. We use the resident/active based predicate `backlog_vip_total>0 & active_vip_size=0 & inflight_vip_total=0` as the main definition (B), and a dispatch-proxy definition (C) as a sanity check.

*Table 12.* **VIP-absence intervals (K=7, W2_phase, W=8).** Mean±std over three seeds.

| Policy | B total (s) | B share | B max (s) | C total (s) | C share | C max (s) |
|---|---|---|---|---|---|---|
| GLOBALFIFO | 0.095±0.005 | 0.00016±0.00001 | 0.0011±0.0001 | 0.479±0.274 | 0.00080±0.00046 | 0.126±0.208 |
| CLIMB | 178.28±4.76 | 0.297±0.008 | 66.7±25.8 | 179.05±4.78 | 0.298±0.008 | 66.7±25.8 |

For CLIMB, the per-seed maximum B gaps are 90.05s, 71.04s, and 39.02s (seeds 101/102/103). These long absence intervals directly inflate the queue component in Eq. (1), explaining mild reversals when $W = K+1$.

# E. CLIMB Implementation Details

This appendix makes CLIMB auditable and reproducible. The main text specifies the contract and mechanism; here we record the implementation semantics needed to reproduce behavior, including the event model, non-preemption rules, corner cases under oversubscription, and a full implementation-aligned pseudocode.

## E.1. State and Invariant

CLIMB maintains per-adapter ingress FIFOs (with arrival timestamps), admitted sets $A^{\mathsf{VIP}}$ and $A^{\mathsf{BG}}$, and per-adapter inflight counters $\mathrm{inflight}[a]$. Let $A_t \triangleq A^{\mathsf{VIP}} \cup A^{\mathsf{BG}}$ and $I_t \triangleq \{a \mid \mathrm{inflight}[a] > 0\}$. We enforce the feasibility invariant on the controller-side resident proxy

$$S_t \triangleq A_t \cup I_t \ \equiv \ \mathrm{resident\_set} \triangleq A^{\mathsf{VIP}} \cup A^{\mathsf{BG}} \cup \{a \mid \mathrm{inflight}[a] > 0\}, \qquad |S_t| \leq K,$$

which matches Eq. (5) in the main text.

## E.2. Event Model and Non-Preemption

The controller runs on each dispatch attempt (i.e., when the router is about to pick the next request). Each step (i) drops idle admitted adapters, (ii) updates active-demand sets for $\mathsf{VIP}$/$\mathsf{BG}$, (iii) admits/rotates adapters subject to $K$, and (iv) dispatches with global FIFO among heads of admitted queues. Deactivation is non-preemptive: an adapter is removed from admission only when it is idle (its ingress queue is empty and it has no in-flight requests). When $\mathsf{VIP}$ oversubscribes the budget, new $\mathsf{BG}$ admissions are paused, but inflight $\mathsf{BG}$ adapters are never preempted and remain counted in $\mathrm{resident\_set}$ until completion.

## E.3. Corner Cases

(i) If $|V_t| \geq K$, the policy enters `bg_paused`: it stops dispatching/admitting $\mathsf{BG}$ (while never preempting inflight $\mathsf{BG}$) and RR-rotates $\mathsf{VIP}$ admissions upon slot release. (ii) If $\mathrm{resident\_set}$ is already full due to inflight adapters, admission may temporarily stall even if backlog exists; progress resumes once an inflight adapter completes. (iii) If no RR-eligible adapter exists (all backlogged adapters are already in $\mathrm{resident\_set}$), the fill step is a no-op.

## E.4. Full Implementation-Aligned Pseudocode

Algorithm 2 provides an implementation-aligned pseudocode of CLIMB, including the resident proxy, the non-preemptive deactivation rule, RR admission/rotation, and GlobalFIFO dispatch among admitted heads.

# F. Design-Space Exploration and Policy Variants

This appendix records the broader design space we explored around CLIMB. The main paper intentionally focuses on a minimal, mechanism-complete story (boundary $\Rightarrow$ knee $\Rightarrow$ mechanism $\Rightarrow$ actionable guard), so we do *not* promote these variants as additional mainline baselines. Instead, the purpose here is pragmatic: (i) to make the mechanism axes explicit and auditable, (ii) to clarify naming/terminology used in exploratory discussions, and (iii) to document a small number of structured alternatives that we implemented to test "what else could have worked" under the same cliff setting.

**Design axes (what we varied).** Across variants, we factor the router-side control into three orthogonal choices: **(1) admission/activation**, i.e., how the policy decides which adapters are eligible to occupy the $K$ LoRA slots (or be considered active); **(2) dispatch**, i.e., how the next request is chosen among eligible adapters/queues; and **(3) optional fairness/stability guardrails**, i.e., additional constraints (deficit accounting, lease/switch budget/cooldown) that shape long-term skew and switching churn. These axes matter differently depending on the regime. The cliff itself is triggered by violating feasibility ($W > K$), which makes residency faults unavoidable unless admission/activation prevents them from entering the engine critical path. Dispatch and guardrails, in contrast, mostly determine *who waits* and *how* waiting manifests (e.g., smooth rotation vs. bursty churn), and therefore affect $\mathsf{BG}$ fairness and stability rather than feasibility.

**Policy catalog and naming convention.** Table 13 lists the variants we implemented, together with a short tag used in plot legends. All names are mechanism-based paper-facing names; internal code identifiers are intentionally omitted.

*Table 13.* Policy variants considered in design-space exploration. **Tag** denotes the short label used in plot legends.

| Policy (paper name) | Tag | Definition |
|---|---|---|
| CLIMB | CLIMB | Hard-gated admission ($K$) with class-aware RR activation and global FIFO dispatch across active adapters; BG admission pauses when VIP backlog exceeds $K$. |
| GATEDRR-GUARD | GDRR-G | Hard-gated admission + per-adapter deficit fairness (DRR) + stability guardrails (lease / switch budget / cooldown). |
| OPENDRR-GUARD | OPENGDRR | No hard gate; keeps per-adapter DRR fairness and stability guardrails. |
| GATERR-GUARD | GATERR-G | Hard-gated admission + stability guardrails; dispatch uses RR among eligible adapters (no deficit). |
| GATEDRR | GATEDRR | Hard-gated admission + per-adapter DRR fairness; stability guardrails disabled. |
| CLASSDRR (SLACK) | CLASSDRR | Same admission as CLIMB; BINDING uses class-level DRR then class-RR; SLACK falls back to CLIMB-style FIFO; optional VIP rescue. |
| URGENCYGATE | URGENCY | Same admission as CLIMB; BG activation picks the most urgent adapter (e.g., max HOL age, $\alpha$queue length); dispatch remains FIFO. |
| SKEWMIXGATE | MIXGATE | Same admission as CLIMB; BG activation mixes RR and urgency with skew-based $\lambda$; dispatch remains FIFO. |
| BGCAP | BGCAP | VIP adapters with backlog/inflight are always activated; BG active set capped by a distinct-adapter cap (`bg_cap`); FIFO dispatch. |
| LRUGATE | LRU | LRU active set with eviction on misses; dispatch keeps locality via clustered queues (`cluster_q`). |
| LOCKGATE | LOCK | Hard-gated admission; once the resident set fills, it is locked until queues drain; FIFO dispatch. |
| GLOBALFIFO | FIFO | No admission control; global FIFO across all adapters. |

To read the table efficiently, it helps to group rows by mechanism axes: (i) GLOBALFIFO as the no-control reference, (ii) feasibility-first variants that enforce a hard gate (e.g., CLIMB, GATEDRR-GUARD, GATERR-GUARD, GATEDRR, LOCKGATE), (iii) variants that keep the same admission skeleton but change how BG is selected for activation (e.g., URGENCYGATE, SKEWMIXGATE), and (iv) structured dispatch alternatives that change the order in which classes/adapters receive service (e.g., CLASSDRR). In particular, CLIMB is the minimal feasibility-first design: it uses a hard gate to respect $K$, a class-aware RR activation rule to decide which adapters become active, and a simple FIFO dispatch among eligible active adapters.

**Why we keep these variants in the appendix (and not the main baseline set).** There are two reasons. First, many variants introduce additional degrees of freedom (e.g., urgency score weights, skew-to-mix rules, class-level deficit semantics), which increases explanation and tuning cost without changing the central feasibility story. Second, some variants are best interpreted as stress points or negative controls: OPENDRR-GUARD removes the hard gate entirely (useful to test whether gating is essential, but not a deployment recommendation), and LOCKGATE intentionally suppresses switching (often introducing throughput confounding), which can be useful for diagnosis but not for claiming Pareto improvements. For the main paper, we therefore prioritize baselines that are (i) widely interpretable, (ii) minimally tuned, and (iii) directly probe the feasibility boundary and the queue-vs-engine mechanism decomposition.

**What this exploration is (and is not) used to support.** We use the catalog in Table 13 to support *interpretability* rather than performance claims: it makes clear which mechanisms are present/absent when we discuss "gating", "dispatch", or "guardrails" in the text. We avoid drawing broad quantitative conclusions from the full variant set because not all variants were run on the full matrix (and some were only used as rapid probes during implementation). The only quantitative comparison we summarize here is a focused structured alternative, CLASSDRR, for which we provide an explicit win/tie/loss summary.

**Focused probe: CLASSDRR vs. CLIMB.** Among the implemented variants, CLASSDRR is the closest structured alternative to CLIMB: it keeps the same admission/activation skeleton as CLIMB, but changes dispatch to allocate service at a *class* granularity before resolving intra-class order. Concretely, CLASSDRR supports two modes: BINDING applies class-level DRR and then RR within the selected class, whereas SLACK falls back to CLIMB-style FIFO dispatch; an optional "VIP rescue" knob can further bias decisions under severe overload. We include CLASSDRR because it represents a natural "more semantic" design direction (more structure than CLIMB) and therefore provides a sanity check on whether

*Table 14.* Win/tie/loss summary for CLASSDRR vs. CLIMB in design-space exploration. Each entry is counted over configuration groups, such as workload, $W$, $\lambda_{\text{VIP}}$, $\lambda_{\text{BG}}$, and vllm_max_lora_rank, using the mean across seeds per group.

| Workload (K=4) | VIP TTFT p99 | BG TTFT p99 | Throughput |
|---|---|---|---|
| W1_main | 2/0/0 | 1/0/1 | 0/1/1 |
| W2_phase | 0/0/3 | 1/0/2 | 0/3/0 |
| Overall | 2/0/3 | 2/0/3 | 0/4/1 |

such additional structure yields robust improvements.

**How to read the W/T/L summary.** Table 14 reports win/tie/loss counts of CLASSDRR relative to CLIMB for three metrics: VIP TTFT p99, BG TTFT p99, and throughput. Each count is computed over configuration groups (e.g., a workload and a small set of run knobs such as $K$ and rank), using the mean across seeds per group (rather than mixing raw samples across seeds). We treat throughput differences within a small tolerance as ties (implementation convention); for latency we use no tie tolerance (any non-zero difference counts as win/loss), while throughput ties use a small tolerance (0.01 rps). This summary is intentionally conservative: it does not hide trade-offs by aggregating metrics into a single score, and it makes it easy to see whether a structured alternative is consistently better across different settings.

**Interpretation and takeaway.** Table 14 shows that CLASSDRR is not a consistent improvement over CLIMB: it wins on VIP tail in some configurations but loses in others, with the overall W/T/L favoring neither a universal replacement nor a clean dominance. This is consistent with our design decision to keep CLIMB as the mainline mechanism: CLIMB already achieves the key feasibility goal (preventing engine-side cliff exposure for VIP by gating), and additional dispatch semantics mainly shift how waiting is distributed, often introducing extra knobs and interpretability cost. We therefore keep CLASSDRR and other variants as appendix material: they document the explored design space and help readers audit mechanism differences, without distracting from the minimal, actionable story in the main paper.

## G. Rank Sweep: Cliff Signature Across LoRA Adapter Sizes

**Motivation and internal control.** To test whether the residency cliff depends on LoRA adapter size, we sweep ranks $r \in \{8, 16, 32, 64, 128\}$ while keeping the serving configuration and the W2 trace fixed. W2 is intentionally piecewise-stationary: only a predefined middle interval expands the active adapter working set (the *cliff interval*), whereas the pre/post intervals remain in the normal fast regime. This structure provides an internal control without introducing additional scheduler baselines.

**Windowed p99 and normalization.** For each rank $r$ and seed $s$, we compute a windowed VIP TTFT p99 time series $p99_{r,s}(t)$ using a fixed time binning shared across all runs. Let $\mathcal{T}_{\text{cliff}}$ denote the predefined cliff interval in W2, and $\mathcal{T}_{\text{ok}} = \mathcal{T} \setminus \mathcal{T}_{\text{cliff}}$ denote the non-cliff portion (pre + post). To remove rank-dependent absolute latency offsets and isolate the on/off cliff signature, we normalize each run by its own non-cliff median

$$b_{r,s} \triangleq \text{median}_{t \in \mathcal{T}_{\text{ok}}} p99_{r,s}(t), \tag{6}$$

and visualize the fold-change as a log-scale heatmap value

$$H_{r,s}(t) \triangleq \log_2 \left( \frac{p99_{r,s}(t)}{b_{r,s}} \right). \tag{7}$$

We aggregate across seeds by taking the median: $H_r(t) = \text{median}_s H_{r,s}(t)$.

**Heatmap visualization and per-rank amplification annotation.** Figure 6 plots the rank–time heatmap of $H_r(t)$. A value of 0 indicates baseline-level tail latency (no cliff), while positive values indicate multiplicative elevation (e.g., $H = 1$ means $2\times$). To summarize the cliff strength per rank without adding additional plots, we optionally annotate each heatmap row with an amplification factor

$$A_r \triangleq \text{median}_s \, \text{median}_{t \in \mathcal{T}_{\text{cliff}}} \left( \frac{p99_{r,s}(t)}{b_{r,s}} \right), \tag{8}$$

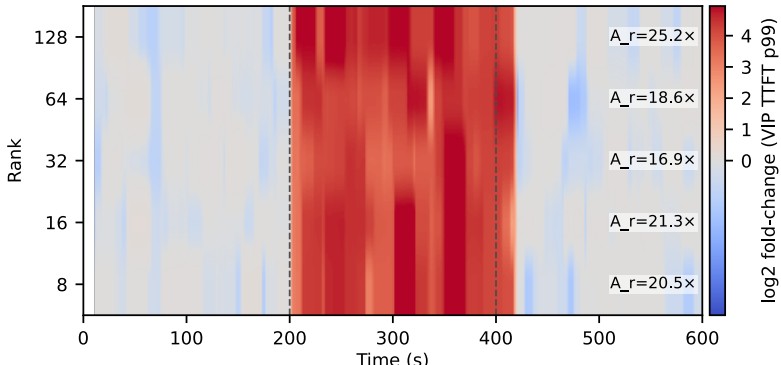

*Figure 6.* Rank sweep under W2: rank–time heatmap of normalized VIP TTFT p99. For each rank $r$, we plot $H_r(t) =$ median$_s$ $\log_2(p99_{r,s}(t)/b_{r,s})$, where $b_{r,s}$ is the run's non-cliff median over $\mathcal{T}_{\text{ok}}$. The shaded region denotes the predefined cliff interval $\mathcal{T}_{\text{cliff}}$ in W2. All ranks exhibit an on/off elevation confined to $\mathcal{T}_{\text{cliff}}$ and return to baseline afterwards. (Optionally, each row is annotated with $A_r$, the median cliff amplification factor.)

*Table 15.* RTX PRO 6000 (96GB), RAM 110GB. Workload: `W2_phase_hol_rps3_p2048_split_M8`, K=4, seeds 101/102/103. Mean $\pm$ std. (seconds; throughput in rps).

| Policy | VIP TTFT p99 (s) | VIP queue p99 (s) | engine p99 (s) | Throughput (rps) | BG TTFT p99 (s) |
|---|---|---|---|---|---|
| GLOBALFIFO | $35.76 \pm 1.86$ | $1.97 \pm 0.07$ | $35.64 \pm 1.67$ | $10.65 \pm 0.23$ | $20.71 \pm 1.75$ |
| CLIMB | $13.94 \pm 3.54$ | $13.88 \pm 3.52$ | $0.08 \pm 0.01$ | $10.65 \pm 0.22$ | $35.66 \pm 9.50$ |

(e.g., shown as "$A_r = \cdot \times$" next to each rank). Across all swept ranks, the elevation is confined to $\mathcal{T}_{\text{cliff}}$ and promptly returns to baseline outside it, indicating that the cliff signature is robust to LoRA rank (adapter size), although its magnitude can vary with rank.

## H. Hardware Robustness: GLOBALFIFO vs. CLIMB on RTX PRO 6000

To test whether the observed residency cliff and the GLOBALFIFO–CLIMB gap depend on a specific GPU, we rerun the same W2 configuration on an RTX PRO 6000 GPU. Table 15 reports the VIP TTFT p99 decomposition (queue vs engine vs total) and throughput for GLOBALFIFO and CLIMB. The RTX PRO 6000 results are consistent with our main observations: GLOBALFIFO exhibits an engine-side cliff (large engine p99), while CLIMB substantially reduces engine p99 and makes the remaining tail primarily attributable to explicit pre-dispatch queueing, without sacrificing throughput.

**Implication: Transfer bottleneck vs. VRAM size.** Although the RTX PRO 6000 provides 96 GB of VRAM—approximately $3\times$ our main RTX 5090 (32 GB; Table 2)—the cliff signature at the same residency budget ($K{=}4$) is qualitatively unchanged. Under GLOBALFIFO, VIP TTFT p99 is still dominated by the *engine* component, whereas CLIMB shifts the tail primarily into explicit pre-dispatch queueing while preserving throughput (Table 15). This supports our interpretation that the cliff is driven by miss-induced adapter loads entering the engine critical path (a host-to-device transfer, e.g., over PCIe), rather than by an outright shortage of physical VRAM. Larger VRAM mainly increases the feasible residency budget $K$ and thus shifts the boundary; once feasibility is violated ($W > K$), such loads can delay token-step iteration boundaries and amplify TTFT tails.

## I. Controller Overhead

This appendix quantifies the overhead of the ingress-side controller used by CLIMB. We report overhead for two practical reasons. First, it rules out a trivial alternative explanation for improved tail latency: a controller could appear "better" simply by throttling admission/dispatch so aggressively that the system processes fewer requests. Second, it demonstrates that the mechanism we advocate is lightweight enough to be deployed in a real serving stack: the decision logic runs on the CPU and should not meaningfully perturb a GPU-bound inference pipeline.

*Table 16.* **Controller overhead (Appendix).** Detailed overhead statistics. Note that `GlobalFIFO` has no controller overhead.

| Policy | Ctrl Time ($\mu$s) | | State | Scaling over $K$ ($\mu$s) | Switch Rate (/s) | L/E (/s) |
|---|---|---|---|---|---|---|
| | Mean | P99 | (KB) | | Mean±Std / P99 | Mean |
| GLOBALFIFO | – | – | – | – | – | – |
| CLIMB | 4.72 | 5.22 | 13.1 | $K4$=3.1, $K8$=4.7, $K16$=7.1 | 0.55±1.0 / 4.0 | 0.55 |

**What counts as "controller overhead".** We separate *control-path compute* from *residency operations*. Control-path compute is the CPU time spent on each controller *tick* to update its internal state and select the next eligible action (e.g., whether to admit/activate an adapter, and which queue to dispatch from). Residency operations (load/evict/swap) are executed by the engine and can be orders of magnitude larger than the control computation; we therefore report them as event rates rather than as per-tick CPU time. This separation is important: CLIMB's goal is to avoid engine-side cliff exposure, not to "compute away" stalls.

**Measurement setup.** We microbenchmark the controller at the **safe anchor** ($W$=8, $K$=8), where residency misses are negligible and engine-side stalls are minimal. Measuring overhead at the anchor keeps the timing stable and avoids conflating controller compute with engine blocking that can occur in the cliff regime. A controller tick is defined as one invocation of the decision routine on the hot path of request handling (specifically, a single call sequence of `update_active_sets` followed by `pick_next_adapter` per scheduler-loop dispatch attempt; arrivals and completions update state consumed by the next decision, and the routine is not run by a fixed-period timer). Thus, the reported tick count measures decision-routine invocations, not a constant-time claim about the internal scans or bookkeeping within those routines. We measure wall-clock time around this routine using a high-resolution timer, after a short warmup. Unless stated otherwise, we exclude file I/O and heavy debug logging from the timed region; the goal is to capture the steady-state compute cost of the logic itself. We report mean and p99 over all ticks in the post-warmup interval, and (when applicable) aggregate across seeds as mean±std.

**Control-path time, state footprint, and throughput at the anchor.** Table 16 reports (i) controller time per tick, (ii) controller state footprint, and (iii) throughput at the safe anchor. We include throughput alongside controller time because overhead must be interpreted in context: a "fast" controller that reduces throughput would still be undesirable. For GLOBALFIFO, the controller is effectively absent, so control-path time/state may be reported as N/A (or as the minimal bookkeeping in the harness). For CLIMB, the measured control-path time is in the microsecond range, while state footprint is small (KB-scale), which is negligible relative to millisecond-scale inference latencies and consistent with "transparent" behavior at the safe anchor. Importantly, the safe-anchor throughput under CLIMB matches the no-control baseline within measurement noise, supporting the claim that CLIMB does not win by throttling when the system is already feasible.

**Additional overhead diagnostics: switching, load/evict, and scaling with $K$.** Beyond per-tick compute, two operational costs are frequently raised by reviewers in residency-control systems: (i) **switching churn** (how often the admitted/active set changes), and (ii) **engine-side residency operations** (adapter load/evict events that can correlate with stalls). To make these costs auditable, we recommend reporting three supplementary statistics:

- **Switch rate** (events/s): the rate at which the controller changes the admitted/active set (or equivalently, activates/deactivates adapters). This is computed from controller-level logs as the delta of `activate_count`/`deactivate_count` over a fixed measurement window.

- **Load+evict rate** (events/s): if the engine exposes explicit load/evict counters, report their average (and optionally p99) rate; otherwise, use the admitted-set switch rate as a conservative proxy and clearly label it as such.

- $K$**-scaling**: re-run the microbenchmark at several $K$ values (e.g., $K \in \{4, 8, 16\}$) under the safe anchor to show how controller time grows with the residency budget. Since CLIMB maintains small per-adapter bookkeeping, we expect near-linear scaling in $K$ rather than superlinear behavior.

These diagnostics are not used to support the main claims, but they help readers reason about deployment cost and tuning risk.

**Interpretation and limits.** The overhead numbers should be interpreted as properties of the *current prototype* and the tested scale (adapter library size and $K$ range). Microsecond-level tick time indicates that the CPU-side control logic is not the bottleneck in our setting; what dominates tail behavior in the cliff regime is engine-side stalling caused by feasibility violations, which CLIMB addresses by gating. That said, overhead can increase in larger deployments (more adapters, richer scoring, more complex fairness semantics), which is precisely why CLIMB keeps the dispatch datapath simple (FIFO) and pushes complexity into a minimal, auditable admission rule. If future variants introduce heavy scoring or additional per-request bookkeeping, we recommend re-running the same microbenchmark and reporting the same summary.

## J. Extended Related Work and Context

This appendix complements Section 6 with broader bibliographic context and additional representative systems. We intentionally include several *orthogonal* but widely-cited directions in modern LLM inference systems (e.g., disaggregated serving, multi-model multiplexing, serverless deployments, offloading, and PEFT variants), to situate CLIMB's focus on *discrete adapter feasibility under continuous batching*.

**Serving architectures beyond monolithic engines: disaggregation and multiplexing.** Recent systems increasingly decouple or multiplex inference phases/resources to improve goodput under SLOs. A representative line disaggregates *prefill* and *decode* across separate GPU pools to reduce phase interference and co-optimize TTFT/TPOT (e.g., Zhong et al., 2024; Patel et al., 2024). Complementarily, multi-model serving work explores spatial-temporal multiplexing and popularity-aware co-location to improve utilization when serving many endpoints concurrently (e.g., Duan et al., 2024). For production-style multi-model marketplaces, token-level or fine-grained pooling/auto-scaling has also been explored (e.g., Xiang et al., 2025). These directions are largely orthogonal to CLIMB: we do not change phase placement or pooling; we instead constrain *which requests enter the engine* to avoid stall cascades triggered by adapter misses.

**Serverless and mobility: cold-start, checkpoint locality, and live migration.** Another adjacent direction targets low-latency *serverless* LLM inference, where cold-start and model loading dominate. Representative work leverages near-GPU storage hierarchies, loading-optimized formats, and live migration of in-flight inference to reduce startup latency (e.g., Fu et al., 2024). While CLIMB focuses on *in-engine* feasibility under a discrete adapter budget, serverless systems highlight a broader design space where placement and mobility decisions interact with tail latency.

**Memory pressure beyond KV paging: heterogeneity-aware allocation and offloading.** Beyond paging-based KV cache management, recent work studies allocator designs for heterogeneous LLM components and layer-specific caching/eviction APIs (e.g., Zhang et al., 2025). When GPU memory is insufficient, heterogeneous/offloading-based inference systems exploit CPU/NVMe to expand effective capacity and/or increase throughput via optimized tensor placement and compression (e.g., Sheng et al., 2023b; Aminabadi et al., 2022). These approaches emphasize memory hierarchy and placement; CLIMB complements them by showing that, under continuous batching, *missing per-request state on the critical path* can induce a sharp tail cliff.

**Speculative decoding and verification as an orthogonal latency lever.** Speculative decoding accelerates autoregressive generation by drafting multiple tokens with a smaller model and verifying with the large model (e.g., Leviathan et al., 2023). Systems such as tree-based speculative inference and parallel verification further operationalize this idea in serving settings (e.g., Miao et al., 2024). This axis is orthogonal to our focus: even with faster decoding, an adapter miss on the critical path can still stall an iteration boundary and amplify tail latency under batching.

**Broader model-serving and GPU-sharing foundations.** Prior to the recent LLM wave, general prediction/model serving systems emphasized batching/caching, multi-tenancy, and robust tail behavior (e.g., Crankshaw et al., 2017; Olston et al., 2017). More recent DNN serving work explores predictability- and SLO-first designs (e.g., Gujarati et al., 2020). At the cluster/GPU layer, fine-grained GPU sharing and scheduling primitives (e.g., Xiao et al., 2018; Shen et al., 2019) provide additional context on why *inference-time multi-tenancy* often requires explicit admission or isolation mechanisms.

**PEFT beyond vanilla LoRA: budgeted and stability-oriented variants.** Many PEFT variants change the size, structure, or effective budget of per-task state, which can interact with multi-tenant serving. Quantization-aware finetuning with LoRA adapters (e.g., Dettmers et al., 2023), adaptive/rank-budgeted LoRA variants (e.g., Zhang et al., 2023), and stability/capacity-oriented LoRA refinements (e.g., Liu et al., 2024; Hayou et al., 2024; Wang et al., 2024) illustrate that "adapter footprint" is

itself a moving target. Other PEFT families (e.g., bias-only finetuning and adapter composition; Zaken et al., 2022; Pfeiffer et al., 2021) and prompt-based tuning (e.g., Liu et al., 2022) reinforce the systems perspective: *many tasks imply many pieces of per-task state*, so feasibility and admission under bounded residency are central in multi-tenant inference.

---

**Algorithm 2** CLIMB implementation-aligned pseudocode (full)

---

**Require:** Budget $K$; RR pointers $p_{\mathsf{VIP}}, p_{\mathsf{BG}}$

1: Maintain per-adapter FIFO queues $Q[a]$
2: Maintain admitted sets $A^{\mathsf{VIP}}$, $A^{\mathsf{BG}}$, a flag `bg_paused`, and inflight counters $\text{inflight}[a]$
    $I_t \triangleq \{a \mid \text{inflight}[a] > 0\}$;   $A_t \triangleq A^{\mathsf{VIP}} \cup A^{\mathsf{BG}}$;   $S_t \triangleq A_t \cup I_t$        $\triangleright$ resident proxy, $|S_t| \leq K$
    $\textsc{NextRRCandidate}(C, p, S_t)$: scan $C$ in RR order from $p$ and return the first $a$ with $|Q[a]| > 0$ and $a \notin S_t$; else return $\emptyset$.
    $\textsc{RRFill}(A, C, K, p, S_t)$: while $|S_t| < K$: $a \leftarrow \textsc{NextRRCandidate}(C, p, S_t)$; if $a = \emptyset$ break; activate $a$; add $a$ to $A$; update $S_t$ (and advance $p$).
3: **loop** on each dispatch attempt
4:     Drop idle admitted adapters: for each $a \in A^{\mathsf{VIP}} \cup A^{\mathsf{BG}}$, if $|Q[a]| = 0$ and $\text{inflight}[a] = 0$, deactivate $a$ and remove it
5:     $W_t \leftarrow \{a \mid |Q[a]| > 0 \ \vee \ \text{inflight}[a] > 0\}$
6:     $V_t \leftarrow \{a \in W_t \mid a \text{ is } \mathsf{VIP}\}$;    $B_t \leftarrow \{a \in W_t \mid a \text{ is } \mathsf{BG}\}$
7:     Update $I_t, A_t, S_t$
8:     **if** $|V_t| \geq K$ **then**                                                           $\triangleright$ `bg_paused`
9:         `bg_paused` $\leftarrow$ true
10:        **for all** $a \in A^{\mathsf{BG}}$ **do**
11:           **if** $\text{inflight}[a] = 0$ **then** deactivate $a$; remove $a$
12:           **end if**
13:        **end for**
14:        Update $A_t, S_t$
15:        $\textsc{RRFill}(A^{\mathsf{VIP}}, V_t, K, p_{\mathsf{VIP}}, S_t)$
16:        $E_t \leftarrow A^{\mathsf{VIP}}$                                                     $\triangleright$ dispatch VIP only
17:     **else**
18:        `bg_paused` $\leftarrow$ false
19:        **for all** $a \in V_t$ **do**
20:           **if** $a \notin S_t$ and $|S_t| < K$ **then** activate $a$; add $a$ to $A^{\mathsf{VIP}}$; update $S_t$
21:           **end if**
22:        **end for**
23:        $\textsc{RRFill}(A^{\mathsf{BG}}, B_t, K, p_{\mathsf{BG}}, S_t)$
24:        $E_t \leftarrow A^{\mathsf{VIP}} \cup A^{\mathsf{BG}}$
25:     **end if**
26:     **if** $E_t = \emptyset$ **then**
27:        **continue**
28:     **end if**
29:     pick $a \in E_t$ whose $Q[a]$ head has the earliest arrival timestamp                 $\triangleright$ global FIFO
30:     dispatch one request from $Q[a]$; $\text{inflight}[a] \mathrel{+}= 1$
31: **end loop**

---

