# OpenReview forum: "CLIMB: Taming the LoRA Residency Cliff in Multi-LoRA Serving"
_ICML.cc/2026/Conference — ICML 2026 regular_

### Official Review · Reviewer_NXTF · 2026-03-09

**Soundness:** 2
**Presentation:** 3
**Significance:** 2
**Originality:** 3
**Overall Recommendation:** 4
**Confidence:** 3

**Summary:**

This paper focuses on the performance degradation incurred by multi-LoRA under limited HBM. If the resident LoRA adapters, controlled by residency budget $K$, are inadequate (less than the requirements, the size of the working set $W$), the fetching stalls the execution and amplifies the P99 tail latency, especially when some requests are delay-sensitive (further small P99 latency). Correspondingly, the authors propose CLIMB, in which it prioritizes those urgent adapters (related to VIP requests with tighter P99). This paper also analyzes the relationship between $K$ and $W$, and further the guided $K$. At last, CLIMB is implemented upon vLLM.

**Compliance With Llm Reviewing Policy:**

Affirmed.

**Final Justification:**

Final recommendation upon rebuttal: weak accept, improved score

Strengths:
The authors study the performance degradation incurred by multi-LoRA, especially when the fetching of LoRA adapters stall existing execution. When the resident LoRA adapters are inadequate, long TTFTs of those blocked requests are unacceptable. Thus, this paper studies the relationship between residency budget and the requirements in an online manner.

Weaknesses:
This paper has shortcomings in answering the following questions.
1. Given HBM, the detailed memory occupation should be listed in revised manuscript.
2. Although $K$=4 is regarded as one explicit bounded-budget operating point used to expose the cliff regime, the best $K$ for diverse devices should be analyzed and evaluated.
3. More general scenario (system with dynamics) should be well studied.

**Key Questions For Authors:**

1. For 32GB HBM (RTX 5090 in the experiments), the detailed memory occupation should be listed (roughly estimation is ok). For example, under limited HBM (32*0.85=27GB, 0.85 utilization already listed in paper), base model occupies 15GB,  the KV occupies xxGBs (the maximum sequence length supported 4k; VIP requests obey 64 length input rps1 output xx tokens, BG requests obey 2k length input rps 3 output xx tokens), and the space left for LoRA adapters is xxGBs (the size per adapter is also needed: 1 VIP adapter with xxMBs and 1 BG adapter with xxMBs). Although the KV occupation varies over time, the minimum space left for LoRA adapters should be estimated.
2. For RTX PRO 6000 (96GB) mentioned in the paper (different HBMs), why the recommended $K$ be 4 again? That is, the authors have to explain ``the cliff signature at the same residency budget (K=4) is qualitatively unchanged'' in the paper.

**Limitations:**

yes

**Strengths And Weaknesses:**

Strengths:
The authors study the performance degradation incurred by multi-LoRA, especially when the fetching of LoRA adapters stall existing execution. When the resident LoRA adapters are inadequate, long TTFTs of those blocked requests are unacceptable. Thus, this paper studies the relationship between residency budget and the requirements in an online manner.

Weaknesses:
This paper has shortcomings in answering the following questions.
1. Dose the size of the LoRA adapter (max. or avg.) affect the recommended value of residency budge $K$? The authors should list the adapter sizes used in the experiments, for both VIP and BG requests. For example, the LoRA adapter for Qwen 2.5 7B is about 200MB. Is it possible to maintain all 8 adapters in HBM? Then, using the workload mentioned in the paper (e.g., VIP requests with prompt length 64, rps 1), the HBM space left for LoRA adapters is about several GBs, which is suitable for at most xx adapters.
2. Does the most suitable $K$ vary over time? Since the KV occupation changes over time, is it possible to prepare more adapters in advance (temporarily enlarge the value of $K$), instead of fixed value (e.g., 4).
3. Does the suitable $K$ change upon various NPU/GPU cards? For 5090 with 32GB HBM, $K=4$ maybe ok. However, for RTX PRO 6000 (96GB) mentioned in the paper, is it possible to set $K$ be 4 again?

---

> ### Author Rebuttal · Authors · 2026-03-28
>
> We thank the reviewer for recognizing the practical importance of limited-memory multi-LoRA serving and for asking for more concrete memory accounting. We agree that the setup and the role of `K` should have been stated more explicitly, and we answer each concern directly below.
>
> **On whether `K` should vary during a run.** In our study `K` is the explicit per-run LoRA-residency cap exposed by the serving configuration / controller (e.g., `max_loras`), rather than a quantity continuously re-inferred from time-varying KV-cache occupancy or instantaneous free memory. In the reported setup, this contract is fixed for a run (`max_loras=4`, `max_model_len=4096`, `gpu_memory_utilization=0.85`). Dynamic memory conditions may affect how a deployment chooses such a budget, but they do not mean that `K` is intended to float during a run in the formulation studied here.
>
> **On whether simply increasing `K` removes the need for CLIMB.** Based on adapter-size accounting alone, some deployments may choose a larger fixed `K`. Our measured `K`-sweep already shows that, for the same workload, increasing `K` shifts the feasibility boundary and can move the system from cliff to safe regime. But choosing `K` is a provisioning / budget decision, whereas CLIMB is the control policy once an explicit bounded residency budget is in place. A larger chosen `K` moves the boundary; CLIMB addresses the complementary question of how to avoid miss/load amplification on the engine critical path under the budget that a deployment actually sets. We therefore do not view “increase `K`” as a contradiction to CLIMB, but as a complementary choice at a different layer.
>
> **On adapter size and memory accounting in the RTX 5090 setup.** We agree that the memory-budget story should be made more concrete. One rank-128 adapter is `~323 MB` (`~0.301 GiB`); thus `4` resident adapters correspond to about `1.20 GiB`, while `8` adapters correspond to about `2.41 GiB`. In the reported RTX 5090 setup, the run-time contract is `gpu_memory_utilization=0.85`, `max_model_len=4096`, and `max_loras=4`, i.e., a configured GPU budget of roughly `32 GB × 0.85 ≈ 27.2 GB`. In the same instrumented mainline run, vLLM reports at initialization a provisioned KV-cache budget of `Available KV cache memory: 8.51 GiB` and `GPU KV cache size: 159,360 tokens`. Increasing `max_loras` from `4` to `8` would add about `1.20 GiB` of static adapter residency under that same fixed budget; this additional residency must come out of the remaining budget that otherwise backs KV-cache and other runtime allocations. We use these as configured / provisioned accounting numbers for the fixed serving contract, not as average or peak runtime usage; equivalently, any momentary unused KV headroom does not automatically become additional LoRA residency during a run. We will report these accounting numbers to make the setup more transparent.
>
> **On the role of the RTX PRO 6000 result.** We agree that the role of the RTX PRO 6000 result should be stated more explicitly. The RTX PRO 6000 result is **not** intended to claim that 96GB hardware should also choose `K=4` as its optimal or recommended setting. Rather, it is a same-budget, same-workload control: we keep `K=4` fixed to test whether the qualitative cliff signature persists under a different hardware envelope. The point of that result is therefore comparative and mechanism-oriented, not a cross-device recommendation of a universal `K`. Larger VRAM can support a larger chosen budget and thus shift the boundary; our Pro 6000 result asks the narrower question of whether, at the same explicit residency budget, the same engine-side cliff signature remains visible.

---

> > ### Author Rebuttal · Reviewer_NXTF · 2026-04-02
> >
> > Thank you for your response. My concerns have been partially resolved. The concerns still remain.
> >
> > As mentioned in the response, RTX 5090 has 27.2GB available memory using gpu_memory_utilization=0.85, in which the model Qwen2.5-7B-Instruct bf16 requires ~15GB, K=4 adapters require 1.2GB, and KV requires 8.5GB. The space left is 27.2 - 15 - 1.2 - 8.5 = 2.5GB. Is it possible to prepare more 4 adapters to have a large K=8 (2.5GB > extra 4 adapters 1.2GB)? Furthermore, the recommended K for RTX PRO 6000 should be discussed.

---

> > > ### Author Response · Authors · 2026-04-03
> > >
> > > Thank you for the follow-up. Under a rough budgeting view, we agree that an RTX 5090 deployment may choose a larger fixed budget such as `K=8`; this is not contradictory to our paper. Our intent is not to argue that RTX 5090 must use `K=4`, or that `K=8` is impossible. Rather, `K=4` in our mainline setting should be read as one explicit bounded-budget operating point used to expose the cliff regime clearly. This is also consistent with our existing evidence: our same-arrival control in Table 9 (Control H, Appendix C) shows that a one-slot change from `K=8` to `K=7` already triggers a cliff-scale jump in VIP TTFT p99 (`0.13 s -> 45.22 s`) while throughput remains essentially unchanged (`10.66 rps`), which is exactly the discrete boundary effect we aim to isolate.
> > >
> > > More broadly, `K` is a provisioning choice made at launch, while the active working set `W_t` is induced by the runtime request stream and varies over time: the operator does not know in advance which adapters will be demanded together by incoming traffic. Thus, even if a deployment provisions a larger fixed `K`, the control problem does not disappear; in general, a static `K` cannot be set equal to `W_t` once and for all. Our contribution therefore addresses a different layer of the problem: given a chosen bounded residency budget, how should the system avoid miss/load amplification on the engine critical path when `W_t` approaches or exceeds that budget? The controlled `W2_phase` workload in the paper is used for reproducibility and mechanism isolation; in deployment, `W_t` is still induced online by the traffic mix rather than scripted at launch.
> > >
> > > The same scope applies to the RTX PRO 6000 result. We do not claim that `K=4` is the recommended setting for 96GB hardware, and a larger fixed `K` may likewise be reasonable in some deployments on RTX PRO 6000. However, we do not provide device-specific recommended `K` values for particular GPU / model / workload combinations in this paper, because that choice depends on the full serving contract (e.g., `max_model_len`, KV-cache provisioning, batching headroom, and workload mix). Our PRO 6000 experiment is intended as a same-`K`, same-workload mechanism control, asking whether the same cliff signature remains visible under a different hardware envelope, rather than as a device-specific tuning recommendation.

---

### Official Review · Reviewer_hjss · 2026-03-11

**Soundness:** 1
**Presentation:** 2
**Significance:** 1
**Originality:** 2
**Overall Recommendation:** 3
**Confidence:** 4

**Summary:**

In multi-tenant LoRA serving, GPU memory is largely occupied by the KV cache, making it infeasible to keep all active LoRA adapters resident on the GPU. Under continuous batching, requests that reference non-resident adapters trigger fetching, which can stall the critical path of batch execution and amplify tail latency for all co-batched requests. When the number of concurrently active adapters exceeds the GPU’s residency capacity, this effect manifests as a sharp latency collapse, termed the LoRA residency cliff. This paper proposes CLIMB, a simple ingress admission control mechanism that enforces the adapter residency constraint explicitly. CLIMB maintains an admitted set of at most K adapters and preferentially batches and dispatches requests whose adapters are already admitted, while holding other requests at ingress. By preventing adapter fetches from entering the engine’s critical path, CLIMB shifts unavoidable delays from engine-side stalls to queueing outside the engine, significantly reducing p99 tail latency for latency-critical requests while preserving throughput.

**Compliance With Llm Reviewing Policy:**

Affirmed.

**Key Questions For Authors:**

1.	The execution model of Algorithm 1 is not fully specified. In particular, when a new request arrives at ingress, is the controller logic (ingress gating, admission, and RR filling) re-evaluated immediately, or is the request simply enqueued into Q[a] with admission decisions deferred until a later dispatch or slot-release event? Clarifying whether Algorithm 1 is executed per-arrival, per-dispatch opportunity, or in a batched/event-driven manner would help readers understand the timing, overhead, and critical-path implications of the proposed queue management.

2.	The paper attributes the limited adapter residency budget K primarily to device memory pressure from the KV cache. However, in modern LLM serving systems the KV cache typically occupies a substantially larger fraction of GPU memory than LoRA adapters. This raises the question of why the design focuses on limiting the number of resident adapters rather than managing or reducing KV cache usage. Could the authors clarify the rationale behind treating K as a fixed constraint while leaving KV cache allocation unchanged? It would also be helpful to discuss whether alternative memory-management strategies (e.g., KV cache compression, eviction, or offloading) could change the effective residency budget and interact with the proposed controller.

**Limitations:**

Yes.

**Strengths And Weaknesses:**

Strengths

A key strength of the paper is its simplicity. The proposed method mitigates the LoRA residency cliff through a minimal ingress control mechanism, avoiding complex runtime scheduling, prefetching, or migration inside the serving engine.
Weaknesses

Weaknesses

1.	 The evaluation in Section 5 treats the adapter residency budget K as a fixed parameter, although in practice K is influenced by both workload-dependent KV cache pressure and CLIMB’s own admission and batching decisions. While the paper attributes the limited number of resident LoRA adapters primarily to KV cache memory usage, KV cache pressure can vary significantly under realistic workloads, potentially changing the effective value of K over time. Moreover, limiting admitted requests at ingress may reduce batch sizes and KV cache usage, which could in turn free memory and allow additional adapters to become resident. This feedback effect between admission control, batching efficiency, and memory availability is not analyzed in the evaluation. As a result, it remains unclear how CLIMB would behave under dynamic memory conditions where the effective adapter residency budget is not fixed.

2.	 While Appendix A specifies concrete parameter choices, the rationale behind fixing these values is not clearly explained. Specifically, the evaluation fixes the workload to a single configuration: one VIP adapter with a prompt length of 64 tokens, seven BG adapters with a prompt length of 2048 tokens, and a maximum generation length of 64 tokens for all requests. It is unclear why these particular values were chosen and whether they reflect representative or favorable conditions for CLIMB.
More importantly, evaluating CLIMB under only a single, highly specific workload configuration makes it difficult to assess the generality of the reported benefits. In realistic deployments, request-length distributions, prompt asymmetry, and VIP/BG class ratios can vary significantly. Without sensitivity analysis across different prompt lengths, generation lengths, or class mixes, it is hard to conclude that CLIMB’s performance gains would hold in more general or diverse settings. Overall, the experimental configuration appears overly restrictive, limiting the strength of the paper’s claims about robustness and real-world applicability.

3.	While CLIMB conceptually shifts latency from engine stalls to ingress queueing when the number of workers exceeds engine parallelism, the paper does not clearly explain the timing and cost of queue management itself. In particular, Figure 2 shows a sudden increase in latency at the W = K + 1 boundary, but it is unclear why queueing overhead becomes significant so abruptly at this point. The paper does not discuss whether ingress queue operations (e.g., enqueue/dequeue, arbitration, and wake-up) lie on the critical path or whether they can be overlapped with engine execution. Without a clear breakdown of these costs, it is difficult to understand whether the observed jump in Figure 2 reflects an inherent limitation of CLIMB or an artifact of the queue management implementation.

---

> ### Author Rebuttal · Authors · 2026-03-28
>
> We thank the reviewer for the careful reading and for pressing on the modeling scope, workload choice, and execution semantics. These boundaries should have been stated more explicitly, and we address each concern directly below.
>
> **On fixed `K` vs. dynamic memory conditions.** In our study, `K` is the explicit per-run adapter-residency cap exposed to the controller / serving configuration (e.g., a fixed cap such as `max_loras`), rather than a quantity continuously re-inferred from instantaneous free memory or KV-cache occupancy. In a broader system sense, admission can interact with batching efficiency, KV usage, and memory availability. Our claim is narrower: dynamic memory conditions may affect how a deployment chooses `K`, but once a run starts the controller contract studied here remains fixed. In this formulation, unused momentary KV headroom does not automatically become additional admissible LoRA residency during a run. Our paper is therefore not claiming joint optimization of KV allocation and adapter residency; rather, it isolates the bounded-residency feasibility boundary that gives rise to the residency cliff under a fixed serving contract.
>
> **On why the paper studies adapter-residency control rather than direct KV management.** KV-cache management is upstream rather than the direct control target here. KV pressure is precisely why practical multi-LoRA serving systems expose only a bounded adapter-residency budget. Our paper focuses on the complementary failure mode that appears once this bounded residency becomes infeasible: when `W > K`, non-resident adapter fetches enter the engine critical path and trigger the residency cliff. In that sense, `K` is a provisioning / budget choice, while CLIMB is the control policy under a given budget; these two are complementary rather than mutually exclusive. When a deployment can afford a larger `K`, the boundary moves accordingly, as our measured `K`-sweep already shows.
>
> **On workload choice and added sensitivity checks.** We chose the main `W2_phase` setting as a diagnostic cliff workload rather than as a claim of representativeness. Within the same `W2_phase` cliff family under the same fixed budget, we added four matched-throughput checks:
> - **Prompt length:** changing `BG prompt_len` from `2048` to `1024` preserves the same VIP-tail ordering (`42.9 -> 11.5 s`, throughput `~10.63–10.64 rps`).
> - **Generation length:** across `max_tokens = 128 / 256 / 512`, CLIMB reduces VIP p99 from `36.5–48.6 s` to `16.7–18.4 s` at matched throughput (`~10.62–10.64 rps`).
> - **Class mix:** across two additional VIP/BG ratio points, CLIMB reduces VIP p99 from `16.9–20.3 s` to `7.85–10.3 s` at matched throughput (`~10.52–10.61 rps`).
> - **Symmetric-prompt negative control:** with equal prompt lengths for VIP and BG, vanilla and CLIMB are nearly identical, with no cliff-scale VIP-tail gap.
>
> Across the asymmetric checks, BG p99 remains higher than vanilla, consistent with the same priority-oriented tradeoff. We therefore present these additions as evidence that the intended VIP-tail benefit is not tied to one single asymmetric setting, not as a claim of universal robustness.
>
> **On the execution model and the `W = K + 1` jump.** The execution model should be stated more explicitly. In our implementation, arrivals are first appended to per-adapter ingress queues, and the controller is evaluated on each dispatch attempt rather than on every arrival. An adapter is idle iff its ingress queue is empty and it has no in-flight requests in the engine; slot release is non-preemptive and occurs only after the adapter becomes idle. This also explains why the jump at `W = K + 1` is a regime change rather than a queue-implementation artifact: once the active working set exceeds the fixed cap `K`, waiting must appear either as engine-side stall amplification (vanilla) or as explicit ingress queueing (CLIMB). In our measured cliff/safe decomposition, vanilla’s `VIP eng p99` is about `38.43 s` at `K=4`, whereas CLIMB reduces `VIP eng p99` to about `0.13 s` while shifting the cost into `VIP queue p99 ≈ 12.97 s`, i.e., queue residence under a binding `K`; at the safe anchor (`W=8, K=8`), controller overhead itself is microsecond-scale (`4.72 us`, p99 `5.22 us`) with no throughput loss. In this measured setting, the large jump is therefore explained by feasibility-induced queueing vs. engine stalls, not by expensive controller bookkeeping.

---

> > ### Author Rebuttal · Reviewer_hjss · 2026-04-03
> >
> > 1. On fixed K vs. dynamic memory conditions. You mention that  "it isolates the bounded-residency feasibility boundary that gives rise to the residency cliff under a fixed serving contract." This assumption is too narrow and restricts the general applicability of the paper.
> >
> > 2. On why the paper studies adapter-residency control rather than direct KV management. How do you explaining evaluating CLIMB under only a single, highly specific workload configuration? This again narrows down the generality of the reported benefits.

---

> > > ### Author Response · Authors · 2026-04-04
> > >
> > > Thank you for the follow-up. We would like to clarify that this does not make the paper’s formulation too narrow for the problem studied here. The sentence quoted from our rebuttal does not introduce a new post-hoc assumption; it restates the problem setting already studied in the paper. In our formulation, `K` is the deployment-chosen, per-run residency budget, and the paper studies admission/control under that fixed bounded contract. This is also how the evaluation is organized: the main paper includes a combined `W/K` sweep, and the supplement includes a same-arrival one-slot control (`K=8` vs. `K=7`), showing that even a one-slot change can flip the system from the safe regime to a cliff regime at essentially unchanged throughput. Our point is therefore not that one universal `K` fits all deployments, but that, once a bounded residency budget is the serving contract for a run, crossing `W > K` creates a discrete feasibility boundary that the controller must handle. Dynamic budget selection is a broader extension, not a prerequisite for the validity of the present formulation.
> > >
> > > On workload generality, our mainline evaluation is indeed centered on a diagnostic cliff family (`W2_phase`), but it is not limited to a single point configuration. The paper/supplement already include `W/K` sweeps, load-matched and liveness controls, an auxiliary `W1` family, a rank sweep, and a second-GPU robustness check. In addition, as stated in our rebuttal, we added four matched-throughput sensitivity checks specifically to address the `prompt/generation/class-mix` concern: prompt length, generation length, class mix, and a symmetric-prompt negative control. To be precise, we do not claim universal robustness across all deployments. Instead, the added prompt-length, generation-length, class-mix, and symmetric-prompt checks provide evidence that the bounded-budget cliff mechanism—and the associated `VIP-tail` benefit—is not tied to one single token configuration.
> > >
> > > We will revise the framing to make this scope boundary more explicit: CLIMB is a control policy for bounded-budget multi-LoRA serving, and is complementary to dynamic KV/adaptive-budget memory management rather than a replacement for it.

---

### Official Review · Reviewer_543g · 2026-03-12

**Soundness:** 3
**Presentation:** 2
**Significance:** 3
**Originality:** 3
**Overall Recommendation:** 5
**Confidence:** 3

**Summary:**

The article presents CLIMB, an ingress controller which aims at reducing
request latency in a multi-LORA serving GPU. This optimization is obtained
by enforcing the queueing of non-resident adapters outside the engine and
prioritizing critical traffic. This controller shows interesting results
where a residency cliff is present i.e. the number of requested adapters
exceeds the number of available slots in the GPU. CLIMB is able to reduce
by a factor 3 the latency of critical requests without reducing the mean
throughput of the system.

**Compliance With Llm Reviewing Policy:**

Affirmed.

**Final Justification:**

The rebuttal addressed my main concerns and reinforced my prior recommendation.

**Key Questions For Authors:**

1. The algorithm 1 is not fully understandable without the text, and may need
   a rework:
   * $A_t$ is not considered a requirement
   * l1: "remove idle adapters from $A_t$", how the idleness is determined? I
     do not think this is discussed in the text

2. The number of runs per experiment is not clear. The mention "mean+std over
   3 seeds" is present, does it means you have multiple runs per seed? Or 3
   runs? Given the high variability of the p99 in fig.2 and tab.1, it may be
   interesting to run it for a higher number of runs.

3. According to the lack of a perspectives section, what do you plan for the
   following of CLIMB? Will it be available in some frameworks/LLM systems?

4. About the feasibility, it is said that in case $|W_t| > K$, the system is
   infeasible and misses are unavoidable, which is undoubtly true. On the
   other hand, the system is feasible if requested adapters can be contained
   in the system. Not sure this is discussed in the article, and maybe that is
   out of the scope, but if between two steps the requested adapters are
   completely different, I suppose there will be an impact on latencies
   because of old adapters to be removed from the engine to load the new
   ones, and thus we can consider the system not feasible for this
   transition? Is this a possible use case? Do you manage it or will it be
   considered in future works?

**Limitations:**

yes

**Strengths And Weaknesses:**

The article presents a lot of strengths. The paper outline is solid and gives
a lot of place to the evaluation and the explanation of the use case, which is
the LORA residency cliff. The evaluation part sounds very robust and clearly
shows how the mechanism answers the issues inherent to the use case.

However, the article is not easy to read due to the technical lexicon.
Fortunately, some of the terms are described in the paper, so the second
reading is sufficient to understand clearly the described mechanisms. Some
acronyms are not obvious to every reader (`BG` for BackGround and `p99` for
the 99th percentile), so clarifications can be written down. Also, the
algorithm lacks information to be completely understandable.

---

> ### Author Rebuttal · Authors · 2026-03-28
>
> **Summary across the three responses:** We clarify that `K` is a fixed per-run adapter-residency budget rather than a runtime-floating quantity, and that our claim is about controlling the residency cliff under a fixed serving contract rather than jointly optimizing KV allocation and adapter residency. To address scope and mechanism questions, we add matched-throughput sensitivity checks across prompt length, generation length, and class mix, plus a symmetric-prompt negative control in which the cliff-scale gap disappears. We also make the execution model and memory accounting explicit, including the queue/engine decomposition and the fixed-budget accounting behind the reported `K`. Together, these clarifications make the claim boundary explicit and reinforce that CLIMB’s benefit is mechanism-consistent under bounded residency, not a claim of universal robustness.
>
> ## Response to Reviewer 543g
> We thank the reviewer for the positive assessment of the paper’s structure and evaluation, and for pointing out places where presentation can be improved. We will expand unclear acronyms such as `BG` and `p99` on first use, and we address the key questions directly below.
>
> **On Algorithm 1 and the definition of idleness.** In our implementation, requests first enter per-adapter ingress FIFOs, and the controller is evaluated on each dispatch attempt rather than on every arrival. An adapter is idle iff its ingress queue is empty and it has no in-flight requests in the engine; slot release is non-preemptive and occurs only after the adapter becomes idle. We will make these operational points explicit so the pseudocode can be read without relying on implementation inference.
>
> **On the number of runs and tail variability.** We clarify that `mean ± std over 3 seeds` means `3 independent end-to-end runs`, one per seed. To reduce uncertainty around tail variability, we added a targeted rerun on the main cliff point in the `W2_phase` family (`W=8`, `K=4`) with 3 additional seeds. These added runs preserve the same qualitative conclusion at matched throughput:
> - **Vanilla:** `VIP TTFT p99 ≈ 45 s`, throughput `≈ 10.57 rps`
> - **CLIMB:** `VIP TTFT p99 ≈ 16 s`, throughput `≈ 10.58 rps`
>
> We present this rerun as additional support for the same mechanism-level conclusion, while agreeing that tail variability in this regime remains non-negligible.
>
> **On follow-up directions and framework integration.** A natural next step is to integrate the same fixed-budget ingress controller into broader serving frameworks built on top of engines such as vLLM. Beyond framework integration, two natural extensions are transition-aware admission for sharper adapter-set turnover and adaptive budget selection layered on top of the same fixed-budget contract studied here.
>
> **On feasibility under adapter-set transitions / churn.** We agree that abrupt turnover of the demanded adapter set can introduce a transition cost even when the instantaneous set size remains within `K`. Under the paper’s instantaneous `|W_t| ≤ K` definition, such a regime is still feasible, but abrupt adapter-set turnover can still incur nontrivial transition latency under churn. Our current feasibility formulation is intended for the bounded active working-set condition itself, rather than as a claim that all within-`K` transitions incur no latency penalty. In this transition/churn regime, CLIMB may still help in the same mechanism-level sense as in the main paper by keeping replacement-induced waiting outside the engine as explicit ingress queueing, rather than letting miss-induced loads amplify engine-side tail latency for co-batched requests. A more explicit transition-aware admission policy is a meaningful future extension, but is beyond the current paper’s scope.

---

> > ### Author Rebuttal · Reviewer_543g · 2026-04-02
> >
> > Thanks for your response. You answered my concerns.

---

### Decision · Program_Chairs · 2026-04-30

**Decision:**

Accept (regular)

**Comment:**

This paper studies an important and practical problem in multi-LoRA serving and proposes a simple method that is technically sound and empirically useful. The idea of controlling admission to avoid the residency cliff is well motivated, and the experiments provide convincing evidence of improved tail latency without hurting throughput.

Some reviewers raised concerns about the fixed-budget assumption, limited workload diversity, and clarity of certain design choices. The authors addressed many of these points in their rebuttal by clarifying the scope of the problem, providing additional analysis, and explaining implementation details more clearly. While some limitations remain, especially around generality and system dynamics, they do not outweigh the strengths of the paper.  In particular, although one reviewer remained concerned about generality, this is a limitation of scope rather than an issue in the core technical contributions.